# Mitophagy in the Pathogenesis of Liver Diseases [note 1]

**DOI:** 10.3390/cells9040831

**Published:** 2020-03-30

**Authors:** Po-Yuan Ke

**Affiliations:** 1Department of Biochemistry & Molecular Biology and Graduate Institute of Biomedical Sciences, College of Medicine, Chang Gung University, Taoyuan 33302, Taiwan; pyke0324@mail.cgu.edu.tw; Tel.: +886-3-211-8800 (ext. 5115); Fax: +886-3-211-8700; 2Liver Research Center, Chang Gung Memorial Hospital, Taoyuan 33305, Taiwan; 3Division of Allergy, Immunology, and Rheumatology, Chang Gung Memorial Hospital, Taoyuan 33305, Taiwan

**Keywords:** autophagy, mitophagy, liver disease, liver injury, hepatitis, steatosis, fibrosis, hepatocellular carcinoma

## Abstract

Autophagy is a catabolic process involving vacuolar sequestration of intracellular components and their targeting to lysosomes for degradation, thus supporting nutrient recycling and energy regeneration. Accumulating evidence indicates that in addition to being a bulk, nonselective degradation mechanism, autophagy may selectively eliminate damaged mitochondria to promote mitochondrial turnover, a process termed “mitophagy”. Mitophagy sequesters dysfunctional mitochondria via ubiquitination and cargo receptor recognition and has emerged as an important event in the regulation of liver physiology. Recent studies have shown that mitophagy may participate in the pathogenesis of various liver diseases, such as liver injury, liver steatosis/fatty liver disease, hepatocellular carcinoma, viral hepatitis, and hepatic fibrosis. This review summarizes the current knowledge on the molecular regulations and functions of mitophagy in liver physiology and the roles of mitophagy in the development of liver-related diseases. Furthermore, the therapeutic implications of targeting hepatic mitophagy to design a new strategy to cure liver diseases are discussed.

## 1. Introduction

Autophagy is a catabolic process that eliminates unwanted intracellular materials via lysosomal degradation to supply nutrients and energy for the maintenance of cellular homeostasis [1,2]. Autophagy is often activated in cells to counteract a variety of stresses, including nutrient starvation, organelle damage, protein misfolding and aggregation, and pathogen infection [3,4]. Loss of control of autophagy regulation is involved in the development of numerous human diseases, such as cancer, metabolic dysfunction, neurodegenerative disorders, and liver-associated diseases [1,2]. Despite its bulk and nonselective degradation mechanism, autophagy has been demonstrated to selectively eliminate damaged organelles to promote organelle regeneration, a process known as “organellophagy” [3,4,5]. Mitophagy is a specific kind of organellophagy responsible for the clearance of damaged mitochondria and thus promotes mitochondrial turnover [6,7]. Improper regulation of mitophagy is implied to participate in the pathogenesis of age-related neurodegenerative and cardiovascular diseases [8,9], metabolic syndrome [10], and tissue injury [11,12]. Very recently, mitophagy was shown to function in the maintenance of hepatic function and to protect the liver from tissue damage via the removal of damaged mitochondria. In addition, deregulation of mitophagy is also implicated in the development of liver-associated diseases, including liver injury, liver steatosis/fatty liver disease, viral hepatitis, hepatic fibrosis, and liver cancer. Thus, a comprehensive understanding of the functional roles of mitophagy will provide an opportunity to identify a new mitophagy modulation-related therapeutic target for the development of a novel and effective strategy for curing and intervening with liver disease. In this review, I summarize the current knowledge on the functional role(s) of mitophagy in the regulation of liver physiology and provide an overview of how mitophagy is altered to prevent and/or promote the development and pathogenesis of liver diseases. Furthermore, I discuss the potential implications of therapeutically targeting hepatic mitophagy for the clinical treatment of liver diseases.

## 2. Autophagy

### 2.1. Discovery of Autophagy and Identification of Autophagy-Related Genes (ATGs)

Autophagy is a “self-eating” process; its name is derived from the Greek words “auto” (self) and “phagy” (eating). During its discovery, autophagy was morphologically characterized by transmission electron microscopy (TEM)-based analysis of double- and single-membraned, vesicle-like, dense structures that sequestered mitochondria and fragments of the endoplasmic reticulum (ER) membrane in different animal tissues [13,14,15,16]. Later observations showed that these dense bodies were related to lysosomal degradation, prompting the discovery of a new cell-autonomous destruction process, which was coined “autophagy” by Christine de Duve, the 1974 Nobel Laureate in Physiology or Medicine, at the Ciba Symposium on Lysosomes in 1963 [17,18]. During the 1970s and 1980s, several stimuli, including hormones and amino acid deprivation, were demonstrated to induce autophagy [19,20,21,22,23,24]. In addition, the signal transduction pathways mediating autophagic processes and pharmacological inhibitors of autophagy, such as 3-methyladenine (3-MA), were characterized and identified [24,25,26,27,28,29,30,31]. Shortly thereafter, in the early 1990s, the concept of the double-membrane structure of autophagic vacuoles derived from the membranes of intracellular organelles, such as endosomes and peroxisomes, was developed [32,33,34,35,36,37,38,39]. The molecular mechanism responsible for autophagy regulation was first unveiled by the isolation and molecular cloning of ATGs by Yoshinori Ohsumi, the 2016 Nobel Laureate in Physiology or Medicine. Ohsumi et al. employed a genetic screen of temperature-sensitive mutants of *Saccharomyces cerevisiae* that were deficient in autophagic degradation and identified approximately 15 ATGs involved in the autophagic process of *Saccharomyces cerevisiae* [40,41,42]. Subsequently, approximately 40 ATGs with homologous functions in autophagy regulation in other eukaryotes and mammals were identified and characterized [43,44,45,46,47,48,49] and were further unified by the autophagy research community [47,48,49].

### 2.2. Three Major Types of Autophagy

To date, three major types of autophagy, namely, macroautophagy, microautophagy, and chaperone-mediated autophagy (CMA), have been identified [50,51]. Among these types, macroautophagy (hereafter referred to as autophagy), which involves a membrane rearrangement process to sequester cytosolic components in autophagic vacuoles and deliver them to lysosomes for degradation, is the most well characterized [51,52]. Various stresses, including nutrient starvation, accumulation of damaged organelles or aggregated proteins, and pathogen infection, have been shown to induce autophagy to eliminate harmful components in cells and maintain cellular homeostasis; thus, autophagy serves as a guardian of human health [53,54]. Accordingly, improper alteration of autophagy has been demonstrated to participate in the pathogenesis of various human diseases and biological processes, such as tumorigenesis, neurodegenerative disorders, infectious diseases, cardiovascular diseases, metabolic syndrome, and aging [55,56,57,58,59,60,61,62,63,64,65,66,67,68,69,70,71]. Microautophagy is an engulfment process that randomly and/or selectively delivers intracellular materials into the lysosomal lumen for degradation through the rearrangement and invagination of the lysosomal membrane into the lumen [72,73,74]. In addition to core ATGs in the autophagic process, the endosomal sorting complexes required for transport (ESCRT) machinery was recently shown to function in the membrane protrusion and scission processes of microautophagy [75,76,77,78]. To date, the molecular process that regulates microautophagy and the physiological importance of microautophagy to human health remain unclear. CMA proceeds through a selective sequestration process involving the recognition of degradative substrates containing the pentapeptide “Lys-Phe-Glu-Arg-Gln″ (KFERQ) motif by a molecular chaperone, heat shock cognate protein of 70 kDa (HSC70), and translocation of these substrates into the lumen of lysosomes through the docking of lysosomal membrane protein 2A (LAMP2A) onto the lysosomal membrane [79,80]. CMA has been shown to be activated by various stimuli, such as nutrient deprivation, metabolic imbalance, oxidative stress, and genotoxicity [81,82,83,84,85,86], and is required for biological processes ranging from energy production, lipid metabolism, gene regulation, immune response control, and cell cycle regulation to aging [82,86,87,88,89,90,91,92,93,94,95,96,97]. Unsurprisingly, deregulation of CMA has also been suggested to contribute to the development of multiple kinds of human diseases [98,99,100,101,102,103,104,105,106,107,108,109,110].

### 2.3. Functional ATGs in the Regulation of Autophagic Process

#### 2.3.1. Membrane Nucleation and Phagophore Formation

The entire process of autophagy relies on the stepwise biogenesis of vacuoles that begins with rearrangement of the membrane for nucleation of the isolation membrane (IM)/phagophore [111,112,113,114]. Different intracellular organelles, such as the ER [115,116], Golgi apparatus [117], mitochondria [118], plasma membrane [119], recycling endosome [120,121], and mitochondria-associated ER membrane [122], supply the membrane source for reconstituting the membranous structure of the IM/phagophore. The cup-shaped IM/phagophore then elongates and matures into a double-membrane autophagosome [123,124,125,126], which subsequently fuses with a lysosome, forming an autolysosome in which the enclosed materials are degraded by lysosomal proteases [125,127,128,129]. In addition, the core ATG complexes and concerted actions of signaling cascades are required for the maturation of autophagic vacuoles (Figure 1) [52,130,131]. In eukaryotic cells, nutrient deprivation often suppresses the activity of mammalian target of rapamycin (mTOR), a serine/threonine protein kinase that regulates cellular metabolism (Figure 1) [132,133]. Suppression of mTOR leads to translocation of the unc-51 like-kinase (ULK) complex, which contains ULK1/2, ATG13, RB1-inducible coiled-coil 1 (RB1CC1, also known as FIP200) and ATG101, from the cytoplasm to a membrane-enclosed compartment derived from the ER (Figure 1) [134,135]. Subsequently, the translocated ULK complex recruits the class III phosphatidylinositol-3-OH kinase (PI(3)K) complex (which contains Vps34/PI3KC3, Vps15, Beclin 1, and ATG14) to the nucleated domain from the ER membrane and activates the class III PI(3)K complex via phosphorylation (Figure 1) [131,136,137,138]. In turn, activation of the class III (PI(3)K) complex leads to the generation of phosphatidylinositol-3-phosphate (PtdIns(3)P) [131,136,137], which induces the recruitment of double-FYVE-containing protein 1 (DFCP1) and WD-repeat domain PtdIns(3)P-interacting (WIPI, the mammalian orthologue of ATG18) family proteins to trigger the reconstitution of an ER-organized omegasome structure (also termed the IM/phagophore) (Figure 1) [131,136,137,139,140]. Moreover, ATG9-mediated vesicle trafficking from the trans-Golgi network (TGN) to the ER and the interaction of ER membrane-bound vacuole membrane protein 1 (VMP1) with the class III PI(3)K complex supply the lipid constituents required for the formation of autophagic vacuoles [141,142,143] and promote the nucleation of the IM/phagophore (Figure 1) [144,145,146].

#### 2.3.2. The Biogenesis of Autophagosome

Two ubiquitin-like (UBL) conjugation cascades are required for the elongation and closure of the IM/phagophore into a mature autophagosome (Figure 1) [147,148,149,150]. The enzymatic activities of ATG7 (E1) and ATG10 (E2) induce the formation of the ATG5-ATG12 conjugate, which further associates with ATG16L to form an ATG5-ATG12-ATG16L trimeric complex (Figure 1) [147,148,151,152,153]. On the other hand, the C-terminal region of ATG8 family proteins (including the microtubule-associated protein 1 light chain 3 [LC3] and gamma-aminobutyric acid receptor-associated protein [GABARAP] subfamilies) are cleaved by ATG4 family proteases to generate ATG8/LC3-I [154,155]. Subsequently, the ATG7 (E1) and ATG3 (E2) enzyme cascade promotes the lipidation (referred to as the conjugation of phosphatidylethanolamine [PE]) of ATG8/LC3-I to form lipidated ATG8/LC3, termed “ATG8-LC3-II” (or lipidated ATG8-LC3) (Figure 1) [154,155,156]. Then, lipidated ATG8/LC3 and the ATG5-ATG12-ATG16L complex cooperate to accomplish autophagosome maturation by promoting the elongation of autophagosomal membranes [157], supporting tethering and membrane fusion between autophagic vacuoles [150,158], and serving as an E3 ubiquitin ligase-like enzyme to promote the formation of lipidated ATG8/LC3 [158,159]. According to recent studies, in addition to these two UBL conjugation complexes, the ESCRT machinery also participates in the closure of IM/phagophore to form autophagosomes [160,161,162].

#### 2.3.3. The Fusion of Autophagosome with Lysosome to Form Autolysosome

Ultimately, an autophagosome fuses with a lysosome, forming an autolysosome that contains acidic proteases to degrade the interior materials. The maturation of autolysosomes requires several biological processes, including cytoskeleton-mediated vesicle trafficking and membrane fusion processes (Figure 1) [123,126,127,129,163,164,165]. The interaction of the autophagosomal membrane-associated small GTPase Ras-related protein 7 (Rab7) with kinesin- and dynactin-linked FYVE and coiled-coil domain-containing 1 (FYCO1) and Rab-interacting lysosomal protein (RILP) allows the transport of autophagosomes along microtubules, facilitating autophagosome-lysosome fusion (Figure 1) [166,167,168,169,170]. On the other hand, histone deacetylase 6 (HDAC6)-triggered remodeling and assembly of the actin cytoskeleton also functions in autolysosome maturation [171,172]. Endosome- and lysosome-associated Rab7 also function in the fusion process between autophagosomal and lysosomal membranes by recruiting pleckstrin homology domain-containing protein family member 1 (PLEKHM1) and the homotypic fusion and protein sorting (HOPS) complex (Figure 1). The interaction between the ATG8/LC3-interacting motif within PLEKHM1 and ATG8/LC3 occurs on the autophagosomal membrane, and the binding of PLEKHM1 to HOPS and the Rab7 complex links autophagosome to the lysosome for fusion [173]. Very recently, this endosomal conversion of phosphatidylinositol-4-phosphate (PtdIns(4)P) to phosphatidylinositol-4,5-bisphosphate (PtdIns(4,5)P_2_) was shown to promote the disassociation of Rab7 and the release of PLEKHM1 from endosomes for autophagosome-lysosome fusion [174]. In addition, the association of the HOPS complex with PI(3)K-associated UV radiation resistance associated (UVRAG) was shown to activate Rab7 and coordinate autophagosome-lysosome fusion (Figure 1) [175]. Moreover, the association of Rubicon and ATG14L with the Beclin 1- PI(3)K complex was indicated to regulate autolysosome maturation [137,176]. Alternatively, the protein complex containing ATG14L, syntaxin 17 (STX17), synaptosome-associated protein 29 (SNAP29), and vesicle-associated membrane protein 8 (VAMP8) was shown to promote the tethering and fusion of membranes for autophagosome-lysosome fusion (Figure 1) [177,178]. In addition to VAMP8, the preferential binding of STX17 to VAMP7A, not VAMP7B, through the regulation of divergent protein kinase domain 2A (DIPK2A) was indicated to control autophagosome-lysosome fusion [179]. Interestingly, the 20S proteasome was recently shown to degrade SNAP29 and STX17 in a ubiquitin-independent manner to control autophagosome-lysosome fusion [180]. ATG8/LC3 proteins were recently shown to play a critical role in autophagosome-lysosome fusion in addition to activating the initial maturation of autophagosomes [181]. Recent studies have demonstrated that Golgi reassembly stacking protein 2 (GORASP2) may promote autophagosome-lysosome fusion by mediating the interaction between LC3 on autophagosomes and LAMP2 on lysosomes [182]. When the autolysosome forms, the encompassed components are degraded by lysosomal proteases in low-pH compartments to support nutrient recycling and energy production.

#### 2.3.4. The Termination of Autophagy

Information regarding the mechanism by which autophagy is terminated and lysosomes are regenerated is still limited. The current model shows that replenishment of nutrients can reactivate mTOR to suppress the induction of autophagy and, coincidently, promote autophagic lysosome reformation (ALR), which terminates the autophagic process [183]. Several other molecules, including clathrin and PtdIns(4,5)P_2_ [184,185], kinesin 1 [186,187,188], and Spinster [189], have been reported to function in ALR. A recent study also implied that the Cullin 3-Kelch-like protein 20 (KLHL20) E3 ubiquitin ligase promotes the degradation of ULK1 and Vps34 complexes to terminate the autophagic process [190].

## 3. Selective Autophagy and Organellophagy

### 3.1. Cargo Recognition in Selective Autophagy

In the past decade, autophagy has no longer been considered only as a nonselective, bulk degradation process. In contrast, autophagy represents a new and specific route to selectively eliminate sequestered organelles and proteins (referred to as cargoes), a process termed “selective autophagy” [3,4,5,191]. Selective autophagy was first observed in the degradation of β-granules in rat pancreatic tissue in the early 1970s [192,193], and the underlying molecular mechanism and regulatory genes were initially delineated and identified in the late 2000s [3,4,5,191]. Before being targeted for degradation by selective autophagy, cargoes are often tagged by polyubiquitination or the binding of an adaptor protein and further recognized by specific cargo receptors, which then deliver the targeted proteins for autophagy by interacting with ATG8/LC3 on the autophagosomal membrane [191,194,195,196,197,198,199]. These cargo receptors, including p62/sequestosome 1 (p62/SQSTM1), neighbor of BRCA1 gene 1 (NBR1), calcium-binding and coiled-coil domain-containing protein 2 (Calcoco2, or NDP52), optineurin (OPTN), and Tax1-binding protein 1 (TAX1BP1) often contain LC3-interacting regions (LIRs) for binding to ATG8/LC3 family proteins and ubiquitin-associated domains for recognizing the ubiquitinated cargoes [191,196,197,199]. Recent studies have identified that in addition to LIR-containing cargo receptors, the ATG8-interacting motifs (AIMs), GABARAP-interacting motifs (GIMs), and ubiquitin-interacting motifs (UIMs) found in several ATGs and other cellular proteins may potentially control the degradation process of selective autophagy [191,198,200,201,202,203,204,205].

### 3.2. Quality Control of Organelle Biogenesis by Selective Autophagy

Selective autophagy has been found to play a functional role in the turnover of damaged organelles, termed “organellophagy” [3,5,206], which protects cells from the consequences of organelle damage and supports the recycling of constituents for the regeneration of the ER, mitochondria, lipid droplets (LDs), peroxisomes, ribosomes, lysosomes, and nuclei (Figure 2) [3,5,206]. To successfully eliminate harmful organelles, several cargo receptors participate in the targeting of damaged organelles for the degradation process of organellophagy [3,5,206]. During mitochondrial turnover by organellophagy (referred to as mitophagy) [6,207], mitochondrial oxidation and depolarization leads to mitochondrial damage [208,209,210,211,212,213,214,215], in turn triggering polyubiquitination of mitochondrial proteins on the outer mitochondrial membrane by the E3 ubiquitin ligase Parkin [213,214,215,216,217,218,219]. Subsequently, Calcoco2/NDP52 and OPTN are recruited for the removal of mitochondria by selective autophagy (Figure 2) [6,220]. In addition to Parkin, other mitochondrial outer membrane (MOM) proteins, such as FUN14 domain-containing 1 (FUNDC1), BCL2/adenovirus E1B 19 kDa protein-interacting protein 3 (BNIP3), BCL2/adenovirus E1B 19 kDa protein-interacting protein 3-like (BNIP3L, or Nix), yeast ATG32, and autophagy/beclin-1-regulator-1 (AMBRA1), also participate in the mitophagy process in a ubiquitin-independent manner (Figure 2) [221,222,223,224,225,226]. Recently, novel cargo receptors for mitophagy, such as prohibitin 2 (PHB2), Toll-interacting protein (Tollip), nitrophenylphosphatase domain and nonneuronal SNAP25-like protein homolog (NIPSNAP) family proteins, and the adenine nucleotide translocator (ANT) complex (Figure 2) [227,228,229,230], were identified. Finally, cargo receptors for mitophagy recruit components of the core autophagy initiation complex, including ULK1, DFCP1 and WIPI/ATG18 family proteins, for autophagosome maturation near the damaged mitochondria [220,231], which involves PtdIns(4,5)P_2_- and F-actin-mediated disassembly of mitoaggregates [232].

For the degradation of damaged peroxisomes by organellophagy (referred to as pexophagy), two protein kinases, yeast Hrr25 and mammalian ataxia-telangiectasia-mutated (ATM), phosphorylate ATG36 in yeast and NBR1 and p62/SQSTM1 in mammals, respectively [233,234,235,236], thus promoting the ability of these mediators to regulate peroxisome turnover by pexophagy (Figure 2). Recent studies demonstrated that the ubiquitination of peroxisomal (PEX) membrane proteins, including PEX5 and the 70-kDa PEX membrane protein (PMP70), may enhance the targeting of damaged peroxisomes by cargo receptors [237,238]. The targeting of damaged ER components for organellophagy, called ER-phagy, requires the functions of several ATGs in yeast, such as ATG39, ATG11, and ATG40 [239], and cargo receptors, including ER-associated reticulon family proteins and family with sequence similarity 134, member B (FAM134B) in mammals (Figure 2) [240,241]. Intriguingly, ATG39 and ATG11 were shown to not only function in ER-phagy but also to regulate nuclear organellophagy, termed “nucleophagy″, to promote the recycling of nuclear components in yeast (Figure 2) [239,242]. Lysophagy is a form of organellophagy that degrades injured lysosomes through the interaction between galectin-3 and LC3 and the p62/SQSTM1-mediated recognition process (Figure 2) [243,244]. Analogously, the removal of other intracellular organelles, including ribosomes (ribophagy) [245,246] and LDs (lipophagy) [247,248], by organellophagy has been shown to regulate cellular homeostasis. However, the specific cargo receptors for organellophagy have not yet been comprehensively identified, the molecular mechanism underlying the regulation of organellophagy is not completely unveiled, and the physiological significance of organellophagy in human health remains to be further studied.

### 3.3. Elimination of Protein Aggregates and Infecting Pathogens by Selective Autophagy

In addition to mediating organelle turnover via organellophagy, selective autophagy also participates in the clearance of protein aggregates and invading pathogens (referred to as aggrephagy and xenophagy, respectively) (Figure 2). For the degradation of protein aggregates by aggrephagy, the p62/SQSTM1 and HDAC6-mediated recruitment of Lys63 (K63)-linked protein ubiquitination determines the specificity of degradative protein aggregates [171,249,250,251]. In addition, NBR1 and autophagy-linked FYVE (ALFY), two cargo receptors, associate with p62/SQSTM1 to target protein aggregates for clearance by aggrephagy (Figure 2) [252,253,254,255]. In addition to removing protein aggregates by aggrephagy, selective autophagy was also found to target specific proteins for degradation. For instance, ferritin heavy and light chains have been shown to be specifically bound by an ATG8/LC3-interacting protein, nuclear receptor coactivator 4 (NCOA4), and targeted for autophagic degradation in a process known as “ferritinophagy″, to precisely maintain the intracellular pool of iron (Figure 2) [256,257] and function in the regulation of erythropoiesis and DNA replication in blood cells [258,259]. To eliminate invading microbes by xenophagy [260,261,262], several cargo receptors, including p62/SQSTM1, NDP52, and OPTN, recognize infecting pathogens and deliver them for degradation (Figure 2) [67,263,264]. Protein kinase-mediated phosphorylation of p62/SQSTM1 and OPTN also serves as a signal for the activation of xenophagy [264,265,266,267]. In summary, selective autophagy plays a critical role in the specific degradation of harmful components to regulate organelle integrity, metabolic homeostasis, and the defense against pathogen infection.

## 4. Mitophagy

### 4.1. Mitochondrial Turnover via Mitophagy

Mitochondria are intracellular factories that not only generate energy in the form of adenosine triphosphate (ATP) via oxidative phosphorylation but also maintain cellular homeostasis by promoting the anabolism of macromolecules and catabolizing metabolic waste [268]. Mitochondria are intracellular organelles that contain the MOM, the mitochondrial inner membrane (MIM), the intermembrane space, and the matrix, which collectively regulate bioenergetics, biosynthesis, and signal transduction [269]. Mitochondria are highly dynamic organelles that undergo cycles of fusion and fission (known as mitochondrial dynamics) to regulate their reshaping, rebuilding, redistribution, and recycling of constituents, thus supporting mitochondrial mass and integrity [269,270]. To control mitochondrial quality in stressed cells, the evolutionarily conserved degradative process of mitophagy is invoked to remove damaged and dysfunctional mitochondria in order to regenerate mitochondria and preserve energy production [6,271].

Current knowledge suggests that mitophagy can be classified into three types according to its physiological roles in cells: basal mitophagy, programmed mitophagy, and stress-induced mitophagy (Figure 3). Less is known about how basal mitophagy (also termed steady-state mitophagy) is regulated in a physiological cellular context, since mitophagy is typically studied and measured under chemical-induced conditions. The observation and monitoring of basal autophagy was instituted by the establishment of mitophagy reporter mice [272,273,274], which are a useful tool for the detection of mitophagy in vivo. These studies in mitophagy reporter mice not only allow us to understand the extent to which mitophagy is differentially activated in various kinds of tissues to routinely regulate mitochondrial regeneration but also suggest that basal mitophagy occurs in tissues to support the metabolic demand [272,273,274,275]. Programmed mitophagy has been shown to be induced to regulate the determination of cell fates, such as erythrocyte differentiation [276,277], cardiomyocyte maturation [278], and stem cell pluripotency [279,280]. In addition, programmed mitophagy was also found to eliminate sperm mitochondria to avoid parental mitochondrial DNA (mtDNA) inheritance [281,282,283,284,285]. Several external stimuli, such as hypoxia, nutrient deprivation, and mitochondrial uncoupling, have been demonstrated to trigger stress-induced mitophagy to promote the clearance of dysfunctional mitochondria [209,224,286,287,288,289]. Deregulation of mitophagy impairs the new synthesis of healthy mitochondria and leads to the accumulation of defective mitochondria, which has been implicated in the pathogenesis of a wide spectrum of human diseases, such as cancer [290,291], neurodegenerative diseases [292,293], cardiovascular diseases [8], tissue injury [11,294,295], metabolic disorders [296], and autoimmune diseases [297]. Thus, modulation of mitophagy has emerged as a potential target for developing new treatment strategies for human diseases.

### 4.2. PINK1/Parkin-Dependent and PINK1/Parkin-Independent Mitophagy

The PTEN-induced putative kinase 1 (PINK1)-dependent protein phosphorylation and Parkin E3 ubiquitin ligase-mediated protein ubiquitination pathway is the most well-studied signaling cascade that regulates mitophagy (Figure 3) [213,214,215,216,298]. Mutations in PINK1 and Parkin have been shown to be associated with mitochondrial dysfunction in the development of neurodegenerative Parkinson’s disease [299,300,301]. Under normal conditions, translated PINK1 is immediately translocated into the IMM through MOM- and MIM-associated translocases; cleaved by protein proteases, matrix processing protease (MPP) and presenilin-associated rhomboid-like protein (PARL) [302,303,304]; and degraded by the E3 ubiquitin ligases UBR1, UBR2, and UBR4 through the N-end rule pathway [305]. When mitochondria are depolarized by external stimuli, impaired mitochondrial import interferes with proteolytic cleavage of PINK1 and suppresses the degradation of PINK [302,303]. Stabilization of PINK1 leads to its accumulation on the MOM and association with the translocase of the MOM (TOM) protein complex [306], thus triggering autophosphorylation and the phosphorylation of ubiquitin at serine (Ser)65 and of Parkin [217,218,298,307,308,309,310,311]. This phosphorylation in turn recruits the E3 ubiquitin ligase Parkin [213,214,215,216,217,218,306] and then promotes the ubiquitination of mitochondrial proteins on the MOM [213,214,215,216,219,306], leading to recognition by specific cargo receptors for the elimination of mitochondria by autophagy [6,213,214,215,216,220,298]. PINK1/Parkin-dependent mitophagy was shown to be required for mitochondrial uncoupler-induced mitophagy and programmed mitophagy [213,214,215,216,224,278,279,280,298,312,313]. 

In addition to PINK1/Parkin-mediated mitophagy, multiple layers of molecular mechanisms have been shown to regulate mitophagy. Several E3 ubiquitin ligases, such as mitochondrial E3 ubiquitin protein ligase 1 (*MUL1*) [282,314,315], SMAD ubiquitination regulatory factor 1 (SMURF1) [316], seven in absentia homolog 1 (SIAH1) [317], Glycoprotein 78 (Gp78) [318,319], Ariadne RBR E3 ubiquitin ligase homolog 1 (ARIH1) [320], and HECT, UBA and WWE domain-containing protein 1 (HUWE1) [226], may specifically ubiquitinate mitochondrial proteins to trigger mitophagy. Intriguingly, the PINK1/Parkin-dependent pathway is not required for the regulation of basal mitophagy [275,321].

### 4.3. Cargo Receptors and Activators of Mitophagy

Several cargo receptors have been identified to function in mitophagy regulation. ATG32, a mitochondria-anchored protein, was the first mitophagy receptor shown to interact with ATG8 and ATG11 and initiate the autophagic degradation of mitochondria to counteract oxidative stress in budding yeast (Figure 3) [287,288,322,323]. Very recently, the mammalian homolog of yeast ATG32, BCL2-like 3 (BCL2-L3), was also proven to function as a mitophagy receptor [324]. In mammals, the BCL2 family protein BNIP3L/Nix was demonstrated to function in mitochondrial turnover by interacting with ATG8/LC3 family proteins during erythrocyte maturation [222,276,277,325,326]. Similarly, BNIP3 was reported to act as a mitophagy receptor for the removal of mitochondria in cardiac myocytes by inducing mitochondrial permeability, cytochrome C release, and mitochondrial fission; interfering with mitochondrial bioenergetics; and binding to LC3 [223,327,328,329,330,331]. BNIP3 and BNIP3L/Nix are also involved in PINK1/Parkin-mediated mitophagy by suppressing the proteolytic cleavage of PINK1 and serving as ubiquitination substrates of Parkin [312,330,332].

p62/SQSTM1 was also shown to activate PINK1/Parkin-dependent mitophagy by bridging the interaction between ubiquitinated voltage-dependent anion-selective channel 1 (VDAC1) on degrading mitochondria and LC3 on autophagosomal membranes [333] and mediating the clustering of depolarized mitochondria [334,335]. In addition to p62/SQSTM1, NBR1 was indicated to be recruited to mitochondria by ubiquitinated BNIP3L/Nix to mediate PINK1/Parkin-dependent mitophagy [332]. FUNDC1 also serves as a cargo receptor for the regulation of mitochondrial dynamics and activation of mitophagy to eliminate mitochondria with hypoxia-induced dysfunction [224,336,337]. The MOM-associated mitofusin 2 (MFN2) is a mitophagy receptor for the quality control of cardiac mitochondria and metabolic maturation of the perinatal heart through phosphorylation by PINK1 and ubiquitination by Parkin [338,339]. OPTN, a causative gene of mitochondrial dysfunction-related amyotrophic lateral sclerosis (ALS) and glaucoma diseases, was shown to participate in PINK1/Parkin-mediated mitophagy by recruiting damaged mitochondria via its LIR [340,341,342]. Unlike wild-type OPTN, the ALS-associated mutant (with a mutation in the ubiquitin-associated domain [E478G]) and LIR mutant of OPTN cannot restore OPTN deficiency-impaired mitochondrial turnover, suggesting the detrimental role of OPTN deficiency in defective mitophagy-linked neurodegenerative diseases [340,342]. 

In addition to mitochondrial proteins located on the MOM, the MIM-associated protein PHB2 was shown to recruit autophagosomal membrane through its LIR domain for PINK/Parkin-dependent mitophagy [228,343]. A recent study showed that PHB2 can destabilize PARL and suppress its proteolytic activity, thus promoting the stability of PINK1 to activate mitophagy [344]. The Parkin-interacting protein AMBRA1 was shown to induce mitophagy by prolonging mitochondrial depolarization and triggering mitochondrial clearance—not by recruiting Parkin to damaged mitochondria [345]. In contrast, other studies indicated that AMBRA1 can drive mitophagy by a LIR motif-mediated interaction with LC3 on autophagosomes and that concerted localization of AMBRA1 is sufficient to activate mitophagy in a PINK1/Parkin-independent manner [346,347].

Recent studies have demonstrated that NIPSNAP1 and NIPSNAP2, two mitochondrial matrix proteins, can translocate to the outer surface of mitochondria and interact with ATG8/LC3 and other mitophagy receptors to play roles in the removal of depolarized mitochondria, which may prevent the occurrence of parkinsonism [230,348]. The ANT complex is another MIM-anchored protein that was recently shown to trigger mitophagy by stabilizing PINK1, and interference with ANT-mediated mitophagy may contribute to mitochondrial abnormalities during the development of cardiomyopathy [229]. The SNARE protein STX17 was similarly indicated to regulate the induction of PINK1/Parkin-dependent mitophagy by interfering with the dephosphorylation ability of a mitochondrial phosphatase, phosphoglycerate mutase family member 5 (PGAM5) [349], and by recruiting the core complex of the autophagy machinery through interaction with ATG14 [350]. Analogously, Rab GDP/GTP exchange factor 1 (RabGEF1)-directed translocation of Rab5 and Rab7 and phosphorylation of Rab7 were also shown to be required for the recruitment of ATG9 vesicles for mitophagy regulation [351,352]. Choline dehydrogenase (CHDH) was demonstrated to translocate to the MOM and form a complex with p62/SQSTM1 and LC3 to activate CCCP-induced Parkin-dependent mitophagy [353]. High-mobility group box 1 (HMGB1), a chromatin-associated protein functioning in nuclear homeostasis, has been reported to regulate mitophagy via the downstream heat shock protein beta-1 (HSPB1, also called HSP27) for mitochondrial quality control [354,355]. 

Although several cargo receptors are indicated to function redundantly in the induction of PINK1/Parkin-dependent mitophagy, recent studies by Youle et al. specifically clarified that NDP52 and OPTN are two detrimental receptors for PINK1/Parkin responsible for recruiting the core machineries of phagophore biogenesis and autophagosome biogenesis [220,356]. In addition, NDP52 was reported to interact with mitochondrial RNA poly(A) polymerase (MTPAP) to form a receptor complex to promote mitochondrial turnover [357]. Collectively, these studies imply that several kinds of cargo receptors function in multiple regulatory mechanisms to control mitophagy. However, new receptors and regulators of mitophagy are still being identified, and the molecular mechanisms underlying the control of mitophagy remain to be investigated and revised, particularly those controlling basal mitophagy and programmed mitophagy—two lesser-studied modes of mitophagy.

### 4.4. Posttranslational Modifications of Mitophagy Regulators

#### 4.4.1. Phosphorylation and Dephosphorylation

Phosphorylation is the most well-studied posttranslational modification (PTM) of mitophagy regulators [358,359,360]. When mitochondria are depolarized and damaged, PINK1 phosphorylates Parkin at Ser65 to stimulate E3 ubiquitin ligase activity [307,308,311,361] and induces the phosphorylation of ubiquitin at Ser65 to establish a feed-forward amplification loop to both promote Parkin ubiquitination and translocation to mitochondria [217,218,307,311,362,363,364,365] and stabilize the assembly of the ubiquitin chain [366]. PINK1 also phosphorylates MFN2 (at threonine [Thr]111 and Ser442) and mitochondrial GTPase 1 (MIRO1) (at Thr298 and Thr299) to activate mitophagy [338,367,368,369]. PINK1-dependent phosphorylation is a positive regulator of mitophagy, and Tank-binding kinase 1 (TBK1) also functions in mitophagy activation by phosphorylating mitophagy receptors. TBK1 phosphorylates OPTN at Ser177 to promote its interaction with ATG8/LC3 proteins [370] and at Ser473 and Ser513 to confer it with the ability to bind to the ubiquitin chain [371,372,373]. In addition, TBK1-mediated autophosphorylation, p62/SQSTM1 phosphorylation at Ser403, Rab7 phosphorylation at Ser72, and other phosphorylation events have been shown to regulate mitophagy [266,374,375,376]. A recent study also noted that the Ca^2+^-binding protein TBC1 domain family member 9 (TBC1D9) may induce mitophagy by activating TBK1 [377]. Casein kinase 2 (CK2) is another kinase involved in the activation of mitophagy by phosphorylating FUNDC1 at Ser13 and phosphorylating translocase of MOM 22 (TOM22) at Ser15 and Thr43 [336,378,379]. On the other hand, adenosine monophosphate-activated protein kinase (AMPK) was demonstrated to phosphorylate ULK1 at Ser555 to promote mitophagy and cell survival under nutrient starvation conditions [380], suggesting a molecular mechanism coupling energy sensing and mitophagy. Additionally, the ULK1 complex was shown to be recruited to depolarized mitochondria [381], presumably through the interaction between FIP200 and NDP52 [220,231]. In addition, ULK1 mediates the phosphorylation of FUNDC1 at Ser17 to promote its interaction with LC3, which facilitates mitophagy induction [225]. Collectively, these studies suggest the promoting role of protein phosphorylation in mitophagy activation.

In contrast to phosphorylation, phosphatase-mediated dephosphorylation was demonstrated to antagonize mitophagy activation. The yeast protein phosphatase 2A-like protein 1 and mammalian PGAM5 were demonstrated to suppress mitophagy by dephosphorylating ATG32, FUNDC1 and dynamin-related protein 1 (DRP1) [349,379,382]. Similarly, phosphatase and tensin homolog (PTEN)-long (PTEN-L), an isoform of PTEN, was recently shown to translocate to the MOM and dephosphorylate Ser65 of ubiquitin to negatively regulate PINK1/Parkin-dependent mitophagy [383,384]. Reciprocally, the expression and mitochondrial localization of PGAM5 were recently shown to be regulated by the PHB2- PARL axis and STX17, respectively, to control mitophagy activation [344,349]. Collectively, these studies indicate that not only phosphorylation but also dephosphorylation represents an antagonistic approach to regulate mitophagy.

#### 4.4.2. Ubiquitination and Deubiquitination

Ubiquitination is a molecular process involving a coordinated cascade of E1 ubiquitin-activating enzymes, E2 ubiquitin-conjugating enzymes, and E3 ubiquitin ligases [385] and is reversibly regulated by deubiquitination through deubiquitinating enzymes (DUBs) [386]. Multiple types of E3 ubiquitin ligases promote the ubiquitination of mitochondrial proteins to tag damaged mitochondria in order to recruit the autophagic machinery for degradation [387]; these proteins include Parkin [365,371,388,389,390], *MUL1* [282,314], SMURF1 [316], SIAH1 [317], Gp78 [318,319], ARIH1 [320], and HUWE1 [226,391]. Parkin-mediated heterotypic ubiquitin chains, such as lysine (K)6, K11, K48, and K63, on mitochondrial proteins of mitophagy are the most often decoded tags [365,371,387,388,389,390]. Several cytoplasmic and mitochondrial proteins have been shown to be ubiquitinated by Parkin during mitochondrial depolarization and to be involved in mitophagy regulation; these proteins include the MOM proteins MFN1, MFN2, TOM20, TOM70, and VDAC family proteins and the cargo receptors of selective autophagy p62/SQSTM1, NDP52, OPTN, and Tax1bp1 [365,371,388,389,390]. MUL1 participates in the ubiquitination of MFN in parallel to PINK1/Parkin-promoted mitochondrial turnover [314]. Additionally, MUL1 has been shown to ubiquitinate ULK1 to positively regulate selenite-induced mitophagy [315]. In addition, MUL1 plays a role redundant with that of Parkin in eliminating parental mitochondria for mouse embryo development [282]. The E3 ubiquitin ligase SMURF1 was reported to target damaged mitochondria for degradation, most likely through the recruitment of autophagosomal membrane to mitochondria via the membrane-binding ability of its C2 domain rather than via ubiquitin ligase activity [316]. SIAH1 was demonstrated to interact with synphilin-1 and translocate to mitochondria in order to promote mitochondrial ubiquitination to regulate PINK1/Parkin-dependent mitophagy [317]; however, the specific substrate that can be ubiquitinated by SIAH1 is unidentified. The E3 ubiquitin ligase activity of Gp78 was shown to be required for the ubiquitination of MFN1 and MFN2 to support mitochondrial turnover during calcium concentration increase-induced and PINK1-independent mitophagy [318,319]. ARIH1 is an E3 ubiquitin ligase widely expressed in numerous cancer cells that has also been demonstrated to ubiquitinate mitochondria to induce mitophagy in a PINK1- but not Parkin-dependent manner, which may contribute to chemotherapeutic resistance during cancer treatment [320]. The HUWE1 E3 ubiquitin ligase promotes PINK1/Parkin-independent mitophagy through multiple processes, including ubiquitination of mitochondria, enhancement of IkB kinase alpha (IKK-alpha)-mediated phosphorylation of AMBRA1 at Ser104, and destabilization of the mitochondrial protein myeloid cell leukemia 1 (MCL1) [226,391]. Another E3 ubiquitin ligase, membrane-associated RING finger protein 5 (MARCH5, also called MITOL), was indicated to negatively regulate hypoxia-inducing mitophagy, in contrast to activating mitophagy, by promoting the degradation of FUNDC1 [392]. In contrast, a recent study showed that MARCH5 may facilitate the ubiquitination of mitochondrial proteins, thus establishing a positive feedback loop to promote PINK1/Parkin-dependent mitophagy [393]. Collectively, these studies imply that ubiquitination serves as an accelerator of mitophagy induction.

Various DUBs, including ubiquitin-specific peptidase (USP)15 [394], USP30 [395,396,397], USP8 [398], USP14 [399], USP33 [400], and USP36 [401], have been demonstrated to regulate mitophagy. USP15 was first shown to counteract PINK1/Parkin-dependent mitophagy by diminishing Parkin-triggered mitochondrial ubiquitination [394]. The mitochondrial DUB USP30 was demonstrated to antagonize PINK1/Parkin-mediated mitochondrial turnover by removing ubiquitin from TOM20 and MIRO [395,396,397]. In addition, recent studies identified two DUBs that suppress PINK1/Parkin-dependent mitophagy: USP33, which acts by removing K6-, K11-, K48-, and K63-linked ubiquitin conjugates on Parkin; and USP36, which acts by downregulating ATG14 [400,401]. In contrast to suppressing mitophagy, USP8-mediated removal of the K6-linked ubiquitin chain from Parkin facilitates efficient recruitment of Parkin to depolarized mitochondria to induce mitophagy [398]. Similarly, USP14 was shown to promote PHB2-mediated mitochondrial membrane rupture in the cellular context of PINK1 and Parkin deficiency [399]. These studies collectively suggest that DUB-dependent deubiquitination may regulate mitophagy through multiple unrevealed and unresolved molecular mechanisms.

#### 4.4.3. Other PTMs

Several other PTMs, such as acetylation, deacetylation, and sumoylation, have been demonstrated to potentially regulate mitophagy. The NAD-dependent protein deacetylase sirtuin-2 (SIRT2) was shown to localize to mitochondria and participate in the regulation of mitophagy via ATG5 deacetylation [402]. Intriguingly, SIRT1 plays an opposite role in mitophagy inhibition by reducing Parkin translocation to mitochondria [403]. In addition to protein deacetylation, the decrease in mitochondrial protein acetylation induced by depletion of the mitochondrial-enriched GCN5-like 1 (*GCN5L1*) gene was demonstrated to induce PINK1/Parkin-independent mitophagy [404]. Another acetyltransferase, the protein N-terminal acetyltransferase A (NatA), was reported to promote yeast mitophagy via ATG32 induction [405]. In addition to protein acetylation and deacetylation, sumoylation of an RNA-binding protein, fused in sarcoma/translocated in sarcoma (FUS), was recently shown to be involved in the regulation of mitophagy in glioma pathogenesis [406]. These results suggest that other PTMs may function in the regulation of mitophagy and remain to be further investigated.

## 5. The Regulation of Liver Function and Liver Diseases by Autophagy

Autophagy has been extensively shown to regulate live physiology in the past few decades. Hepatic autophagy was first shown to promote glycogen degradation to balance liver metabolism and protect hepatocytes from liver atrophy [407,408,409]. This regulation of hepatic metabolism by autophagic degradation is triggered by deprivation of nutrients [21,410,411] and altered levels of metabolites [410,412,413]. Similarly, autophagy also functions in the turnover of intracellular RNA and proteins [414], as well as organelles, such as Mallory-Denk bodies (MDBs) [415,416,417], ER [418,419], mitochondria [420,421,422,423], peroxisome [424,425,426,427,428,429,430,431,432] and LDs [88,89,247,248,433,434,435,436]. In addition, autophagy also participates in the protection of liver cells from damage [437,438], the balancing of the intracellular levels of amino acids and iron [439,440], and the regeneration of transplanted liver tissues [441]. Thus, these studies collectively underlie that autophagy plays a protective role in maintaining the balance of liver metabolism and in protecting liver cells from injury.

In addition to the physiological role of autophagy in liver, modulation of autophagy was shown to participate in the pathogenesis of liver diseases. Autophagy has been demonstrated to eliminate the aggregate of alpha (1)-antitrypsin Z mutant in the liver of patients with alpha (1)-antitrypsin deficiency to prevent liver against cell death [442,443,444,445]. Deregulated LDs catabolism by abnormal autophagy was also reported to be involved in the development of liver steatosis and fatty liver diseases [446,447,448,449]. Additionally, impaired autophagy was evidenced to be correlated with the pathogenesis of liver cancer. The genetic ablation of ATG5 and ATG7 has been reported to spontaneously induce the development of liver cancer through the accumulation of p62/SQSTM1-driven nuclear factor erythroid-2-related factor 2 (Nrf2) and induction of antioxidant responses [450,451]. The hyperphosphorylation of p62/SQSTM1 and downstream Nrf2-antioxidant axis was recently shown to promote metabolic reprogramming to enhance the chemoresistance of hepatocellular carcinoma (HCC) cells [452], and to enhance the diethylnitrosamine (DEN) carcinogenic activity for HCC development [453]. Moreover, autophagy has consistently been shown to be activated in liver cells infected with hepatitis viruses, including hepatitis B virus [454,455,456], and hepatitis C virus [457,458,459,460]. Based on these findings, alterations in autophagy may be involved in the development of liver diseases and represent as a specific target for the diagnosis and treatment of liver diseases.

## 6. The Physiological Role(s) of Mitophagy in the Liver

### 6.1. The Discovery of Mitochondrial Autophagy in the Liver

The liver was the first tissue leading to the discovery of mitochondrial autophagy; in the early 1960s, Ashford et al. observed mitochondria sequestered within autophagic vacuoles associated with lysosomes in glucagon-perfused rat hepatocytes [14]. Additionally, mitochondria were the most common intracellular organelles initially observed within autophagic vacuoles in the late 1950s and early 1960s [13,14,15,16]. Soon thereafter, mitochondrial autophagy was observed in rat hepatic tissues under different kinds of stimuli, including exposure to the detergent Triton WR-1339 [461], fasting [461], exposure to glucagon [20,411,462], and amino acid deprivation^25^. The TEM-based ultrastructural and biochemical fractionation analyses in these studies suggested that a large portion of fragmented and dysfunctional mitochondria are engulfed by autophagic vacuoles, in which lysosomal acidic proteases degrade the enclosed materials [20,461,462], suggesting that hepatic autophagy may function in mitochondrial turnover. Later, the concept of selective degradation of organelles, including mitochondria, by autophagic process, emerged from biochemical and morphological studies on liver tissues [23,411,463,464,465,466,467,468,469,470,471,472]. In addition, autophagic vacuoles in hepatic tissues were shown to contain mitochondrial enzymes [35,36,473,474,475]. These studies collectively imply that mitochondrial autophagy may promote mitochondrial degeneration in the liver.

### 6.2. The Functional Role(s) of Mitophagy in Liver Physiology

The procedure for the specific detection of mitophagy was initially established by immunogold TEM-mediated labeling of mitochondrial enzymes within autophagic vacuoles and fluorescent probe-based tracking of mitochondrial engulfment by autophagy [475,476]. Mitochondrial degradation by autophagy was first demonstrated in the liver tissues from patients with alpha (1)-antitrypsin (α1-AT) deficiency, a chronic liver disease (Table 1) [444,477,478], suggesting that autophagic degradation of mitochondria may be activated to counteract the mitochondrial injury induced by α1-AT deficiency. In addition, failure of autophagic mitochondrial degradation was shown to be associated with age-dependent accumulation of 8-oxo-2′-deoxyguanosine (8-OHdG) in mtDNA in the rat liver, and activated mitochondrial autophagy was demonstrated to degrade injured mitochondria and rescue older cells from 8-OHdG-mtDNA accumulation (Table 1) [423,479]. Komatsu et al. first demonstrated that ATG7 deficiency led to the accumulation of fragmented and deformed mitochondria in mouse liver tissue (Table 1) [480]. Mitochondrial degradation by autophagy was further shown to be induced by nutrient starvation in order to remove depolarized mitochondria from the mouse liver via the pre-autophagosomal structure (PAS)-mediated nucleation process [421]. Loss of BNIP3, a cargo receptor for mitophagy in the mouse liver, was reported to induce an increase in the mitochondrial mass and the accumulation of abnormal mitochondria (Table 1) [481], suggesting the functional role of BNIP3-dependent mitophagy in the maintenance of mitochondrial integrity. These studies may imply a critical role of basal mitophagy in mitochondrial quality control. On the other hand, interference with hepatic autophagy by ischemia/reperfusion (I/R) and anoxia/reoxygenation (A/R) was indicated to trigger mitochondrial dysfunction by enhancing mitochondrial permeability transition (MPT) (Table 1) [482], and aging can aggravate I/R-induced liver injury via Parkin-mediated mitophagy (Table 1) [483]. In addition, sequestration of mitochondria within autophagic vacuoles was shown to be correlated with acute liver cell damage in patients with anorexia nervosa (Table 1) [437]. Thus, these studies imply that stress-induced mitophagy is pivotal to mitochondrial homeostasis [482].

Later, mitochondrial degradation by autophagy was shown to be activated during the remodeling of primary rat hepatocytes through the MPT, suggesting that mitophagy underlies hepatic remodeling in response to stimuli (Table 1) [420]. Consistent with this observation, mitophagy was demonstrated to be triggered by ethanol in mouse liver tissues (Table 1) [446], and Parkin was shown to be required for ethanol-induced mitophagy (Table 1) [484,485,486], suggesting that mitophagy may mitigate ethanol-induced hepatotoxicity. In addition, efavirenz, the specific reverse transcriptase inhibitor used to treat HIV, was reported to induce mitophagy to rescue mitochondrial dysfunction, thus circumventing drug-induced liver damage (Table 1) [487,488]. Additionally, dynamin 1-like-dependent mitochondrial fragmentation was shown to potently activate mitophagy in liver cells to mediate cadmium-induced hepatotoxicity (Table 1) [489]. Moreover, mitophagy was indicated to be involved in the removal of dysfunctional mitochondria to overcome acetaminophen-induced liver injury (Table 1) [490,491], most likely through a PINK1/Parkin-independent pathway (Table 1) [492,493]. In addition, hepatic mitophagy was reported to be activated by AMPK signaling and to inhibit NACHT, LRR and PYD domains-containing protein 3 (NLRP3) inflammasome activation to diminish acetaminophen-triggered hepatotoxicity (Table 1) [494,495]. Collectively, these findings indicate that mitophagy fundamentally promotes mitochondrial turnover in liver cells and acts as a guardian to prevent hepatic cells from injury.

## 7. Role(s) of Mitophagy in the Development of Liver Diseases

### 7.1. Liver Injury

The association of mitophagy and/or mitochondrial autophagy with liver injury was first discovered in earlier studies showing that α1-AT deficiency leads to mitochondrial dysfunction by interfering with autophagic degradation of mitochondria (Table 2) [444,477,478]. α1-AT is a serum glycoprotein that functions as an inhibitor of destructive neutrophil proteases [496,497]. Several naturally occurring mutants of α1-AT [498], including the S variant (in which the glutamic acid [Glu] at residue 264 [Glu264] is mutated to valine [Val]) and the Z variant (in which Glu264 is mutated to lysine [Lys]), have been identified and shown to participate in the pathogenesis of human diseases, such as chronic liver-associated diseases [499,500,501]. Homozygosity for the Z variant of α1-AT (α1-ATZ) leads to a conformational change promoting α1-AT polymerization and the formation of inclusion bodies that are retained in the ER, thus impairing α1-AT secretion into the bloodstream and causing liver damage [502,503,504]. Numerous studies have demonstrated that autophagy may function in the elimination of α1-ATZ protein aggregates to prevent hepatic injury [442,443,505,506,507]. Interference with the autophagic pathway by genetic knockout of ATGs, such as ATG5, was demonstrated to inhibit α1-ATZ degradation in yeast cells [505,508]. Reciprocally, enhancement of autophagy by rapamycin and other autophagy-enhancing drugs was proven to promote the clearance of α1-ATZ to restore α1-AT deficiency-induced liver damage [509,510,511,512]. α1-ATZ aggregation was reported to induce mitochondrial dysfunction in the liver tissues of patients with α1-AT deficiency (Table 2) [444]. Later, mitochondrial injury-associated mitophagy was shown to be induced in liver tissues of patients with α1-AT deficiency and in the livers of α1-ATZ transgenic mice (Table 2) [477,478]. In addition, defects in mitophagy were shown to enhance the MPT and induce mitochondrial dysfunction in the setting of I/R- and A/R-induced hepatic injury (Table 2) [482]. I/R-induced downregulation of SIRT1 was demonstrated to reduce MFN2 deacetylation (at the Lys655 and Lys662 residues) and impair mitochondrial autophagy, thus inducing mitochondrial dysfunction and liver injury (Table 2) [513,514]. Overexpression of SIRT1 by adenoviral gene delivery and enhancement of SIRT1 activity by a pharmacological activator, SIRT1702, were shown to restore mitochondrial autophagy and to suppress mitochondrial abnormalities and cell death after I/R (Table 2) [513]. Consistent with these studies, recent studies indicated that aging may reduce Parkin and ATG5 expression, interfering with the mitophagy process and exacerbating hepatic I/R injury (Table 2) [483], and that Parkin deficiency can aggravate hepatic I/R injury by suppressing mitophagy-mediated mitochondrial turnover (Table 2) [515]. Heme oxygenase-1 (HO-1) was recently demonstrated to restore hepatic I/R-impaired mitophagy by activating PGAM5 signaling, which protected against hepatocellular damage during I/R injury (Table 2) [516]. In contrast, microRNA 330-3p (miR-330-3p) was reported to elevate hepatic I/R injury by suppressing PGAM5-mediated mitophagy (Table 2) [517]. Moreover, mitophagy was demonstrated to remove dysfunctional mitochondria to protect liver cells from hepatotoxicity induced by drugs such as acetaminophen [490,491], efavirenz [487], and cadmium [489] (Table 2). In addition, mitophagy was proven to prevent both acute ethanol-triggered hepatotoxicity in mice (Table 2) [446,518] and polyethylenimine-induced liver damage [519]. Collectively, these studies imply that mitophagy may act to protect liver cells from different kinds of liver injury and serve as a therapeutic target for the design of a new therapy for liver injury.

### 7.2. Steatosis and Fatty Liver Diseases

Numerous studies have shown that autophagy may balance lipid metabolism by catabolizing LDs [88,89,248,433,436,520,521] and regulating LD biogenesis [247,522]. Thus, autophagy has been indicated to prevent the development of fatty liver, and elevation of autophagic activity is considered a strategy to be exploited for developing therapeutics to treat fatty liver disease [435,523,524,525]. The induction of mitochondrial damage in alcoholic liver disease was first indicated during the 1970s and 1980s by the increased levels of giant and rounded mitochondria in liver biopsies from patients with alcoholic liver disease and in the fatty liver tissue of ethanol-treated mice (Table 3) [526,527,528]. The relationship between mitophagy and hepatic steatosis was first observed in an earlier study showing that the expression of BNIP3, a mitophagy receptor, is induced by fasting and that loss of BNIP3 triggers hepatic steatosis and promotes its transition into steatohepatitis in mice (Table 3) [481]. Additionally, loss of BNIP3 was shown to result in reduced mitochondrial integrity and increased lipogenesis in the livers of BNIP3-null mice [481], suggesting the critical role of mitophagy in regulating normal liver metabolism and preventing hepatic steatosis. Moreover, the induction of hepatic steatosis in rats by ethanol was demonstrated to dramatically activate mitophagy by elevating PINK1 expression on mitochondria to eliminate damaged mitochondria (Table 3) [529]. In addition, ethanol-activated PINK1-dependent mitophagy was shown to be strongly correlated with the mitochondrial expression of Parkin and the level of the indicator of oxidative mtDNA damage, 8-OHdG, in the rat model of ethanol-induced hepatic steatosis (Table 3) [530,531]. The functional role of Parkin-dependent mitophagy in alcohol-induced hepatic steatosis was further evidenced by decreased mitophagy, mitochondrial respiration, and cytochrome c oxidase activity (Table 3) [484,486]. Based on these studies, mitophagy plays a protective role in combating alcohol-induced mitochondrial dysfunction, hepatic steatosis, and hepatic injury, indicating a promising opportunity to develop feasible therapeutic and preventative strategies for alcoholic fatty liver disease. In support of this idea, enhancement of mitophagy by quercetin, a naturally occurring flavonoid, was shown to alleviate ethanol-triggered mitochondrial damage [532].

Despite the role of mitophagy in alcoholic hepatic steatosis, accumulating evidence also indicates that mitophagy may be involved in the development of nonalcoholic fatty liver disease (NAFLD), which encompasses a wide spectrum of progressive diseases ranging from nonalcoholic fatty liver (NAFL) to nonalcoholic steatohepatitis (NASH) to liver cirrhosis and, consequently, to hepatocellular carcinoma (HCC) and represents a major public health burden associated with a modern lifestyle [540]. Mitochondrial abnormalities in the liver tissues of patients with NAFLD were first demonstrated by the formation of megamitochondria containing linear crystalline inclusions (Table 3) [533]. Later, diet-induced NAFLD in mice was shown to block hepatic autophagy and lead to oxidative stress and mitochondrial dysfunction (Table 3) [534]. Additionally, diet-induced NAFLD was indicated to reduce thyroid hormone-induced mitophagy in mice, suggesting the role of insufficient mitophagy in NAFLD-induced abnormal mitochondrial homeostasis and hepatic injury (Table 3) [535]. In addition, induction of NAFLD in HepG2 cells treated with oleic acid (OA) was shown to activate mitophagy through expression of the p53-dependent damage-regulated autophagy modulator (DRAM) to promote hepatocyte apoptosis (Table 3) [536]. According to another study conducted in mice, diet-induced NAFLD suppresses the completion of mitophagy and leads to the accumulation of mitophagy intermediates, e.g., megamitochondria that contain p62/SQSTM1, Kelch-like ECH-associated protein (Keap1), and ubiquitin (Table 3) [537]. Mitochondrial stasis induced by simultaneous deletion of two dynamin-related GTPases for division (DRP1) and fusion (OPA1) in the mouse liver rescues the mitochondrial integrity and liver function by restoring Parkin-independent mitophagy mediated by p62/SQSTM1/Keap1/RING-box protein 1 (RBX1) (Table 3) [537]. Moreover, increased degradation of mitochondrial oxidative phosphorylation subunits was suggested to contribute to the suppression of mitophagy and induction of mitochondrial defects in hepatocytes from mice with NAFLD (Table 3) [538]. On the other hand, impairment of mitophagy was shown to activate the NLRP3 inflammasome to promote the progression of NAFL to NASH [539]. These studies collectively indicate the pathological role of deregulated mitophagy in the development of NAFLD and suggest the therapeutic potential of exploiting mitophagy enhancement for the design of a new curative therapy for NAFLD. Consistent with this idea, several studies have reported that restoration of mitophagy may ameliorate the progression of NAFLD [541,542,543,544].

### 7.3. Liver Cancer

The alteration of mitochondrial homeostasis in liver cancer was first discovered in the 1950s [545,546]. In subsequent studies, TEM ultrastructural analyses of subcellular compartments in liver specimens from patients with liver cancer and in liver tissues from mice with safrole-induced HCC further indicated that the loss of mitochondrial integrity is associated with the initiation of hepatocarcinogenesis (Table 4) [547,548]. Concanavalin A (ConA), a lectin that functions in the activation of acute hepatic inflammation, was shown to suppress hepatoma cell growth in vitro through BNIP3-mediated mitophagy and to inhibit the formation of liver tumor nodules in vivo (Table 4) [549,550]. Adriamycin, a chemotherapeutic drug commonly used to treat HCC, and curcumin, an extract of *Curcuma longa*, were also reported to activate mitophagy to promote the apoptosis of hepatoma and HepG2 cells (Table 4) [551,552]. Moreover, melatonin was shown to trigger mitophagy to increase the sensitivity of human HCC cells to the cytotoxic effects of sorafenib, a kinase inhibitor approved for the treatment of liver cancer (Table 4) [553]. Consistent with these studies, DRAM-mediated mitophagy was implied to promote the apoptosis of HCC cells (Table 4) [554]. According to a recent study, FUNDC1-mediated mitophagy may suppress the initial development of HCC in mice by interfering with inflammasome activation (Table 4) [555]. Although it can play a protective role in liver tumorigenesis, mitophagy could be an initiator and/or accelerator of hepatocarcinogenesis. Mitochondrial fission, which promotes cell survival via reactive oxygen species (ROS) production and is positively correlated with a poor prognosis for patients, was frequently reported to be elevated in liver specimens from patients with HCC (Table 4) [556]. Mitophagy was also shown to attenuate p53 activity to support the maintenance of hepatic cancer stem cells (CSCs), thus promoting the development of liver cancer (Table 4) [557]. When mitophagy is suppressed in cells, PINK1-mediated phosphorylation of p53 at Ser392 in the mitochondria induces nuclear translocation of p53 and prevents the expression of NANOG gene, a homeobox transcription factor that mediates the maintenance of CSC stemness and self-renewal ability, thus decreasing the hepatic CSC population (Table 4) [557]. Notably, FUNDC1 was reported to accumulate in the liver tissues of patients with HCC (Table 4) [555], suggesting that FUNDC1-induced mitophagy may contribute to the pathogenesis of late-stage HCC. To date, whether mitophagy is altered and how mitophagy participates in the development of cholangiocarcinoma remain largely unknown, although p62/SQSTM1-mediated mitophagy and regulation of mitochondrial dynamics were shown to sensitize cholangiocarcinoma cells to the cytotoxic effects of the chemotherapeutic drug cisplatin [558]. Collectively, these studies indicate that mitophagy plays a multifaceted role in preventing and/or contributing to liver cancers. However, further investigations are required to comprehensively delineate the functional role(s) of mitophagy in the progression of liver cancer. In perspective, mitophagy modulation is a valuable strategy to be exploited for the development of a new therapy for liver cancer.

### 7.4. Viral Hepatitis

In the past decade, autophagy has been widely demonstrated to function in the life cycles of hepatitis viruses, including hepatitis B virus (HBV) and hepatitis C virus (HCV) [457,458,459,460,559,560,561,562]. HCV infection was shown to induce autophagy to promote the replication of viral RNA [458,559,560,561,563,564], translation of incoming viral RNA [459], and assembly of infectious virions [565,566,567,568]. In addition, HCV-activated autophagy was reported to suppress the innate antiviral response [459,460,562,569], protect infected hepatocytes from cell death [570], and promote LD catabolism in infected liver cells [459,571]. HCV infection was also reported to induce PINK1/Parkin-dependent mitophagy, in which mitophagosomes contribute to mitochondrial injury associated with chronic HCV infection (Table 5) [572,573]. Moreover, HCV infection was shown to trigger DRP1 phosphorylation at Ser616 and translocation of DRP1 to mitochondria, thus inducing mitochondrial fission and mitophagy [574]. Virus-activated mitophagy was further indicated to attenuate apoptosis and promote persistent viral infection (Table 5) [574]. Consistent with this finding, the HCV nonstructural protein 5A (NS5A) was reported to disrupt mitochondrial dynamics, by concomitantly increasing ROS production and triggering mitophagy (Table 5) [575]. In contrast, the HCV core protein was demonstrated to suppress mitophagy by interacting with Parkin and interrupting its translocation to mitochondria to sustain HCV-induced mitochondrial injury in infected liver cells (Table 5) [576]. Collectively, these studies suggest that HCV may subvert the removal of damaged mitochondria by mitophagy to prevent the death of infected cells and maintain viral persistence.

The role(s) of autophagy in HBV infection were first revealed by an earlier study showing that heterozygous deletion of Beclin 1, an ATG protein in the PI(3)K complex, suppresses autophagy in liver cells, thus promoting HBV-associated premalignant lesions [578]. Later, HBV-induced autophagy mediated through an increase in PI(3)K enzyme activity was further demonstrated in an in vitro cell culture model and in liver tissues of HBV transgenic mice and was shown to promote viral DNA replication [454,455,456]. HBV x protein (HBx) and HBV small surface protein (SHBs) were shown to induce autophagy to support viral replication [579,580]. Autophagy and the autophagic machinery were indicated to participate not only in viral genome replication [454,455,456,579,580,581,582], but also in the secretion of infectious virions by infected cells [583,584], viral envelopment [580], virus-induced secretion of cytokines [585,586], the suppression of HBV-associated tumorigenesis [587], and the degradation of infecting virions [588,589]. In contrast, HBx was demonstrated to impair autophagic flux by suppressing lysosomal function to contribute to the development of HBV-associated HCC [590]. HBV has been shown to induce Parkin-mediated mitophagy though DRP1-mediated mitochondrial fission to attenuate the apoptosis of infected cells (Table 5) [577]. Additionally, thyroid hormone was reported to trigger mitophagy to protect hepatocytes from HBx-induced carcinogenesis [591]. Moreover, HBx was recently shown to enhance Parkin-dependent mitophagy through the Lon peptidase in hepatocytes under starvation conditions [592]. These studies imply the functional roles of mitophagy in HBV-host interactions.

### 7.5. Other Liver Diseases

An impairment of mitophagy was shown to potentially participate in carbon tetrachloride (CCl_4_)-induced hepatic fibrosis in the mouse liver, and melatonin was indicated to restore mitophagy and protect hepatocytes from hepatic fibrosis (Table 6) [593]. In addition, another study showed that interference with T-cell immunoglobulin domain and mucin domain-4 (TIM-4) in Kupffer cells may suppress PINK1/Parkin-dependent mitophagy to mitigate CCl_4_-induced hepatic fibrosis in mice (Table 6) [594]. Mitophagy was also reported to be induced by increased ROS production to promote PM2.5-induced hepatic stellate cells (HSCs) activation and hepatic fibrosis (Table 6) [595]. In contrast, inhibition of mitophagy was shown to promote inflammation in HSCs during acute liver injury (Table 6) [596]. Very recently, activation of DRP1-mediated mitochondrial fission and suppression of FUNDC1-dependent mitophagy were shown to promote DNA-dependent protein kinase catalytic subunit (DNA-PKcs)-induced alcoholic liver disease (Table 6) [597].

Mitophagy was also shown to control the fatty acids metabolism. Genetic studies in mice showed that the deletion of BNIP3 in liver triggers lipogenesis by elevating the expressions of lipogenic enzymes and decreasing the β-oxidation of fatty acids, thus increasing the mitochondrial mass and upregulating hepatic respiration [481]. On the other hand, the β-oxidation of fatty acids was shown to be enhanced through inducing the gene expression of carnitine palmitoyltransferase 1α (CPT1α), a rate-limiting enzyme required for fatty acid oxidation by thyroid hormone-activated mitophagy in liver cells [598]. Similarly, CPT1α expression was recently shown to be increased and accompanied by the increased expressions of mitophagy-related genes, such as BNIP3L and Parkin in REDD1 (regulated in development and DNA damage response-1) knockout mice with NAFLD [599]. These studies together concluded that hepatic mitophagy may integrate lipid metabolism through increasing fatty acids β-oxidation.

It was also evinced that mitophagy is regulated by hepatic insulin resistance. Insulin has been shown to interfere with autophagy by suppressing the expressions of ATGs, such as ATG12 and Vps34, in cultured hepatocytes [600]. Moreover, starvation- and glucagon-induced mitophagy are repressed by insulin resistance in cultured hepatocytes, which are induced by the long-term exposure to insulin [600]. Intriguingly, gene knockout of Parkin in mice does not increase insulin resistance and obesity, presumably due to an impairment in intestinal lipid absorption [601], and a decrease in ER stress and the activation of AMPK [602]. Loss of FUNDC1-mediated mitophagy was demonstrated to increase insulin resistance through inducing adipose tissue-associated macrophage infiltration and hyperactivation of mitogen-activated protein kinase 8 (MAPK8, also named JNK1) [603]. Thus, these studies imply that mitophagy is not only regulated by hepatic insulin resistance, mitophagy but also functions in the insulin resistance-associated metabolic disorder. Collectively, these studies indicate that mitophagy may participate in the pathogenesis of other liver diseases. However, the exact role(s) of mitophagy in the development of these liver diseases are still enigmatic and required further investigation.

## 8. Translation of Mitophagy to the Control of Liver Diseases

Because autophagy and mitophagy have been extensively shown to be modulated in the pathogenesis of liver diseases, the knowledge of mitophagy regulation might conceivably be translated into the clinic for the diagnosis and treatment of liver diseases. Mitophagy and mitophagy-related genes are differentially regulated in liver diseases, and thus the monitoring of changes in mitophagy and the mitophagy regulators might represent a promising strategy to diagnose liver diseases. For example, p62/SQSTM1 accumulates in the damaged mitochondria in the liver during NAFLD development [537], and the increased expression of FUNDC1 supports the tumorigenesis of HCC [555]; thus, these proteins represent potential biomarkers for diagnosing liver diseases. On the other hand, the mitophagy-associated alteration in the activation of the inflammasome and downstream interleukin secretion might represent another diagnostic marker for the early detection of NAFLD and HCC [539,555]. Notably, conclusions on the role(s) of mitophagy in liver diseases were drawn from animal experiments, rather than from clinical samples obtained from patients with liver diseases. Further investigations of the modulation of these potential mitophagy-related biomarkers in cohorts of patients with liver diseases are urgently needed to confirm the reliability and feasibility of these new methods designed for diagnosing liver diseases.

An increase in hepatic mitophagy has been shown to protect liver cells from damage and to prevent the progression of liver diseases, thus suggesting that the precise activation of mitophagy potentially represents a rational strategy to interfere with the development of liver diseases. For example, the induction of mitophagy has been widely shown to prevent liver injury caused by drugs and ethanol [446,487,489,490,491,518,519]; therefore, a study aiming to search for and identify the highly potent inducers of mitophagy for clinical used as treatments for liver injury would be valuable. Several enhancers of mitophagy that trigger or restore mitophagy have been identified, such as p62/SQSTM1-mediated inducer (PMI) [604], urolithin A [605], quercetin [532], and ivermectin [376]. However, researchers have not confirmed whether these inducers are able to be safely utilized for treating liver diseases because extensive clinical trials of these drugs in patients with liver diseases have not been successfully completed. Questions still exist regarding potential side effects of these mitophagy inducers after administration for curing liver disease. Very recently, an increase in mitophagy mediated by the induction of the MPT has been shown to unexpectedly induce hepatic cell death in response to I/R injury and to shorten the lifespan [606]. Therefore, further studies are needed to determine whether manipulations of mitophagy represent a druggable target for systematically curing liver diseases. The comprehensive knowledge of the role(s) of mitophagy in liver diseases will allow us to overcome obstacles of applying the therapeutic modulation of mitophagy in the clinic. In addition to mouse genetic models, studies of the alterations in mitophagy in liver specimen and the identification of the clinically relevant correlation between altered mitophagy and the pathogenesis of liver diseases will help us to search for more specific molecular targets, such as a particular cargo receptor or regulator of mitophagy that is dispensable in general mitophagy, and develop a new therapeutic strategy to treat liver diseases.

## 9. Conclusions and Perspectives

Mitophagy plays a pivotal role in the elimination of dysfunctional mitochondria to promote mitochondrial turnover and maintain mitochondrial integrity, thus physiologically regulating liver function. On the other hand, accumulating evidence implies the extensive modulation of mitophagy in various types of liver diseases. Induction of mitophagy has historically been believed to protect liver cells from damage and injury and to serve as a guardian to prevent the development of liver diseases. Accordingly, the enhancement and/or restoration of hepatic mitophagy seems to be a promising strategy to exploit for the development of new therapeutics for liver diseases. On the other hand, mitophagy is subverted to promote the pathogenesis of liver diseases, suggesting that proper suppression of mitophagy could be harnessed to alleviate the progression of liver diseases. However, the implications of manipulating hepatic mitophagy to treat liver diseases remain largely uncertain, most likely due to the functional role(s) of mitophagy in liver diseases, which are controversial and under debate. Additionally, there are large discrepancies among different studies, and most of the conclusions on mitophagy and liver diseases are experimental model- and disease stage-dependent, thus impeding the comprehensive understanding of the functions of mitophagy in liver diseases. However, further investigations are required to unveil the detailed role(s) of mitophagy in the development of liver diseases and to improve knowledge on the clinical relevance of autophagy in different stages of liver disease progression. Most importantly, innovative and integrative biomedical research approaches should be incorporated to identify molecular targets of mitophagy, such as a specific cargo receptor, that can be translated into the design of a safe, feasible, and effective clinical therapeutic strategy for liver diseases.

## Figures and Tables

**Figure 1 cells-09-00831-f001:**
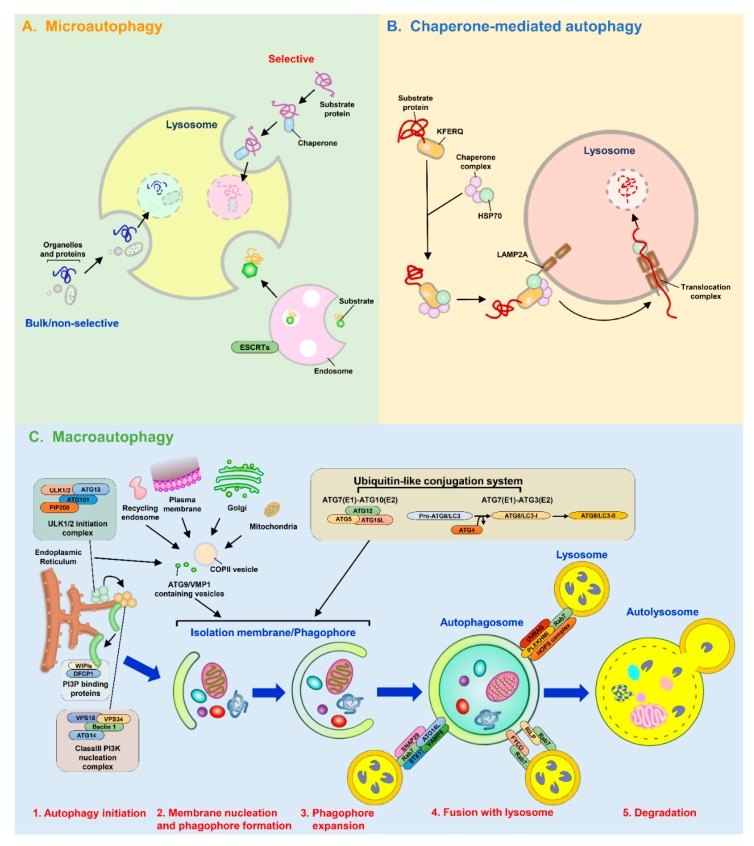
Overview of the molecular mechanisms regulating autophagy. Autophagy is classified into three major types: microautophagy, chaperone-mediated autophagy (CMA), and macroautophagy. (**A**) Microautophagy is a dynamic lysosomal membrane process that directly enwraps and deliver the intracellular portions to the lumen of lysosomes for degradation. (**B**) CMA is a lysosomal degradation process that involves the recognition of proteins containing the KFERQ-like motif by HSC70 and transport into the lysosomal lumen via the interaction with LAMP2A. (**C**) Macroautophagy (referred to here as autophagy) is a degradation pathway that promotes membrane rearrangement to generate vacuoles to engulf the cytosolic components targeted for degradation within lysosome. Two distinct metabolic sensors, AMPK and the mTOR complex, control the initiation of autophagy. Nutrient starvation in cells triggers the suppression of mTOR by AMPK1and the translocation of the ULK1/2 complex (ULK1/2, ATG13, RB1-inducible coiled-coil 1 (RB1CC1, also known as FIP200) and ATG101) to the autophagy initiation site. Subsequently, the ULK1/2 complex leads to the recruitment and activation of the class III phosphatidylinositol-3-OH kinase (class III-PI3K complex, including Vps34/PI3KC3, Vps15, Beclin 1, and ATG14) to synthesize PtdIn(3)P. The generated PtdIn(3)P then recruits DFCP1 and WIPI family proteins to the ER-associated membrane compartment to induce the formation of the isolation membrane (IM)/phagophore. In addition to the ER, other organelles, such as the plasma membrane, mitochondria, Golgi apparatus, and recycling endosome, also supply membrane resources required for the membrane nucleation and the formation of phagophore in the initial step of autophagy. ATG9, VMP1, and coated protein complex-associated vesicles are involved in the trafficking of membrane resources for the formation of the phagophore. The expansion and elongation of the phagophore to form enclosed autophagosomes rely on two ubiquitin-like (UBL) conjugation systems. First, ATG12 is conjugated to ATG5 via ATG7 (ubiquitin activating enzyme 1, E1) and ATG10 (ubiquitin conjugation enzyme 2, E2), yielding an ATG5-ATG12 conjugate that binds to ATG16L to form an ATG5-ATG12-ATG16L complex. Second, ATG8/LC3 family proteins are post-translationally processed by a cysteine protease ATG4 to form ATG8/LC3-I. Then, ATG7 (E1) and ATG3 (E2) enzyme cascades mediate the covalent linkage of ATG8/LC3-I to PE to form the lipidated form of LC3 (ATG8/LC3-PE, also known as ATG8/LC3-II). The enclosed autophagosomes fuse with lysosomes to form mature autolysosomes, which eliminate the engulfed materials. The interactions of RAB) with FYCO1 and RILP regulate the fusion of autophagosomes with lysosomes. In addition, several protein-protein interactions and the assembly of protein complexes, including the interaction between UVRAG and RAB7, the association of PLEKHM1 with the HOPS complex, and the formation of a protein complex containing ATG14L, STX17, SNAP29, and VAMP8, also regulate autophagosome-lysosome fusion.

**Figure 2 cells-09-00831-f002:**
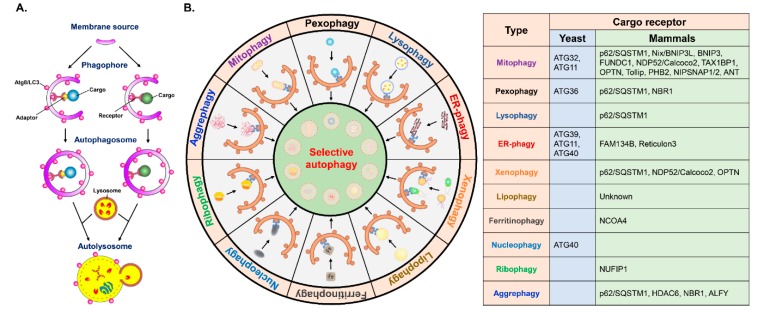
Regulation of selective autophagy by cargo receptors. (**A**) Selective autophagy is an elimination process that involves the targeting of cargoes to the autophagy machinery by the specific receptor proteins that contain an ATG8/LC3-interacting regions (LIRs) for the interaction with ATG8/LC3 located on the membrane of the IM/phagophore. Another type of selective autophagy is mediated by the interactions between adaptor proteins with cargo receptors and ATG8/LC3. The IM/phagophore encloses mature autophagosomes that fuse with lysosomes to form autolysosomes, in which the engulfed cargoes are degraded. Ubiquitination of cargo and/or the binding to additional adaptor proteins mediate the recognition of cargoes and cargo receptors. (**B**) Selective autophagy participates in the degradation of damaged organelles and aggregated proteins. Several mammalian and yeast cargo receptors have been identified to eliminate the dysfunctional mitochondria, damaged lysosomes, injured peroxisomes, stressed ER and infecting pathogens by selective autophagy (mitophagy, lysophagy, pexophagy, ER-phagy, and xenophagy, respectively). Moreover, other forms of cargoes, including LDs (lipophagy), ferritin (ferritinophagy), nucleus (nucleophagy), ribosome (ribophagy), protein aggregate (aggrephagy), are degraded by selective autophagy through a process mediated by identified and unknown cargo receptors. The yeast and mammalian cargo receptors responsible for each type of selective autophagy are listed in the table.

**Figure 3 cells-09-00831-f003:**
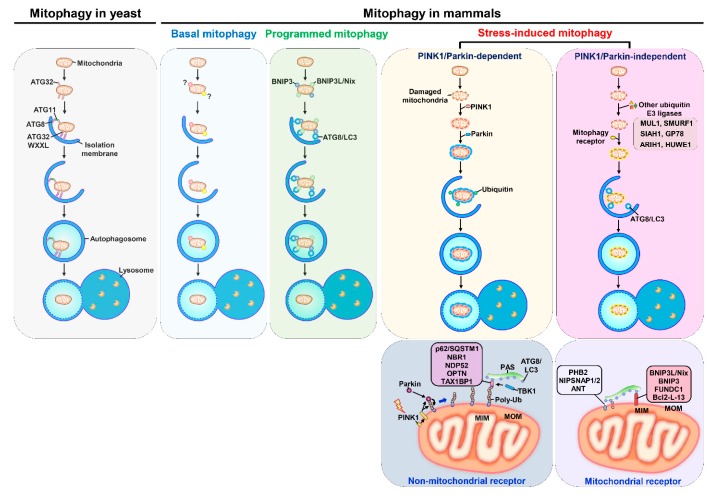
Different types of mitophagy in yeast and mammals. In yeast cells, the mitochondrial outer membrane-associated protein ATG32 represents a receptor involved in mitophagy that interacts with ATG8 located on IM/phagophore through the WXXL-like motif. Another adaptor protein, ATG11, may mediate the interaction between ATG32 and ATG8 to facilitate mitophagy initiation. At least three types of mitophagy have been in mammals, including basal mitophagy, programmed mitophagy, and stress-induced mitophagy. Basal mitophagy has been shown to be activated in the tissues with large metabolic demands, but the specific receptor for basal mitophagy remains unclear. The cargo receptors of programmed mitophagy, including BNIP3 and BNIP3L/Nix, function in mitochondrial turnover for the maturation of erythrocytes and cardiomyocytes, and elimination of the parental mitochondrial genome during fertilization. Stress-induced mitophagy is classified into PINK1/Parkin-dependent mitophagy and PINK1/Parkin-independent mitophagy. For the activation of PINK1/Parkin-dependent mitophagy, mitochondrial depolarization triggers the suppression PINK1 degradation and leads to PINK1 accumulation on the MOM of damaged mitochondria. In turn, PINK1 phosphorylates ubiquitin and Parkin at serine residue 65 and then promotes the mitochondrial translocation of Parkin and mitochondrial ubiquitination by Parkin. On the other hand, several ubiquitin E3 ligases, including MUL1, SMURF1, SIAH1, GP78, ARIH1, and HUWE1, may promote the ubiquitination of mitochondrial proteins to activate PINK1/Parkin-independent mitophagy. Several non-mitochondrial cargo receptors, including p62/SQSTM1, Calcoco2 (also known as NDP52), OPTN, NBR1, and TAX1BP1, are responsible for PINK1/Parkin-dependent mitophagy. The phosphorylation of cargo receptors by tank-binding kinase 1, such as OPTN and p62/SQSTM1, facilitate the recognition process of mitophagy. Several MOM- and MIM-associated proteins, such as BNIP3, BNIP3L/Nix, FUNDC1, BCL2-L3, PHB2, ANT complex, and NIPSNAP1 and 2 have been identified as cargo receptors for mitophagy that directly bind to ATG8/LC3 located on the IM/phagophore.

**Table 1 cells-09-00831-t001:** Summary of the roles of mitophagy in liver physiology.

Experimental Model	Characteristics of Mitophagy	Function of Mitophagy	References
1. Liver specimens from patients with an alpha (1)-antitrypsin (α1-AT) deficiency 2. Liver tissues of α1-AT Z variant (α1-ATZ) transgenic mice	Electron micrographs showed autophagic vacuoles that engulfed mitochondria in the liver tissues of human patients and α1-ATZ transgenic mice	The sequestration of deformed mitochondria associated with α1-AT deficiency-mediated chronic liver diseases	[444,477,478]
Rat liver tissues (ageing)	1. Age-dependent accumulation of 8-hydroxy-2′-deoxyguanosine (8-OHdG) in the mitochondrial DNA (mtDNA) in the liver tissues of aged mice 2. Age-dependent decrease in cytochrome C oxidase activity in the liver tissues of aged mice	Age-dependent loss of mitophagy activity	[423,479]
Liver tissues of wild type and ATG7 knockout mice	Electron micrographs showed autophagic vacuoles that engulfed mitochondria in the liver tissues of ATG7 knockout mice	Degradation of damaged mitochondria by mitophagy	[480]
Liver tissues of GFP-LC3 transgenic mice	1. Sequestration of GFP-LC3-labeled mitochondria in the liver tissues of nutrient-starved GFP-LC3 transgenic mice 2. Engulfment of GFP-LC3-labeled mtDNA in the liver tissues of nutrient-starved GFP-LC3 transgenic mice	Degradation of dysfunctional mitochondria by starvation-induced mitophagy	[421]
Liver tissues and primary hepatocytes from wild type and BNIP3 knockout mice	1. Immunofluorescence staining for Hsp60, a mitochondrial matrix protein, was observed in primary hepatocytes isolated from wild type and BNIP3-null mice 2. Immunofluorescence staining of MitoTracker-labeled mitochondria in primary hepatocytes isolated from wild type and BNIP3-null mice	Reduced mitochondrial turnover and increased mitochondrial mass following the loss of BNIP3-dependent mitophagy	[481]
Rat liver tissues (ischemia/reperfusion (I/R) and anoxia/reoxygenation (A/R) treatments)	Sequestration of GFP-LC3-labeled mitochondria in the liver tissues of I/R- and A/R-treated mice	1. Degradation of dysfunctional mitochondria by mitophagy 2. Ageing aggravated the I/R-induced impairment in Parkin-dependent mitophagy	[482,483]
Liver specimens from patient with acute liver damage induced by anorexia nervosa	Electron micrographs showed autophagic vacuoles that engulfed mitochondria in liver tissues from human patients	Degradation of damaged mitochondria by starvation-induced mitophagy	[437]
Primary rat hepatocytes	1. Electron micrographs showed autophagic vacuoles in the dedifferentiation-induced remodeling of rat hepatocytes 2. Immunofluorescence staining of MitoTracker-labeled mitochondria in the dedifferentiation-induced remodeling of rat hepatocytes	Remodeling of hepatocytes by mitophagy	[420]
1. Liver tissues from wild type and GFP-LC3 transgenic mice (ethanol treatment) 2. Isolated primary mouse hepatocytes (ethanol treatment)	1. Electron micrographs showed autophagic vacuoles that engulfed mitochondria in the liver tissues of ethanol-treated mice 2. Engulfment of MitoTracker-labeled mitochondria by GFP-LC3-labeled autophagic vacuoles	1. Degradation of damaged mitochondria by ethanol-induced mitophagy 2. Protection against ethanol-induced liver injury by Parkin-dependent mitophagy	[446,484,485,486]
1. Hepatocytes, Hep3B cell line 2. Primary human hepatocytes (treated with the antiretroviral drug efavirenz)	1. Electron micrographs showed autophagic vacuoles that sequestered enlarged mitochondria in the efavirenz-treated liver cells 2. Immunofluorescence staining of MitoTracker-labeled mitochondria and LysoTracker-labeled lysosomes in the efavirenz-treated liver cells	1. Degradation of damaged mitochondria by efavirenz-induced mitophagy 2. Mitophagy protected against efavirenz-induced hepatic injury	[487,488]
The human normal liver cell line, L02 (cadmium treatment)	1. Electron micrographs showed autophagic vacuoles that engulfed mitochondria in the cadmium-treated liver cells 2. Immunofluorescence staining revealed the co-localization of TOM20-labeled mitochondria with GFP-LC3 in the cadmium-treated liver cells 3. Immunofluorescence staining of MitoTracker- and chloromethyl-X-rosamine (CMXRos)-labeled mitochondria in the cadmium-treated liver cells	1. Degradation of damaged mitochondria by cadmium-induced mitophagy 2. Mitophagy protected against cadmium-induced hepatotoxicity	[489]
Liver tissues from wild type and GFP-LC3 transgenic mice (acetaminophen treatment)	1. Electron micrographs showed autophagic vacuoles that engulfed mitochondria in the acetaminophen-treated mouse hepatocytes 2. Immunofluorescence staining of MitoTracker-labeled mitochondria in the acetaminophen-treated mouse hepatocytes 3. Degradation of mitochondrial proteins in the acetaminophen-treated mouse hepatocytes	1. Degradation of damaged mitochondria by acetaminophen-induced PINK1/Parkin-dependent mitophagy 2. Protection against acetaminophen-induced liver injury by PINK1/Parkin-dependent mitophagy 3. Activation of acetaminophen-induced mitophagy through AMPK activation and the suppression of inflammasome activation	[489,490,491,492,493,494,495]

**Table 2 cells-09-00831-t002:** Summary of the roles of mitophagy in liver injury.

Experimental Model	Characteristics of Mitophagy	Function of Mitophagy	References
1. Liver specimens from patients with an α1-AT deficiency 2. Liver tissues from α1-ATZ transgenic mice	Electron micrographs showed autophagic vacuoles that engulfed mitochondria in the liver tissues of human patients and α1-ATZ transgenic mice	Sequestration of deformed mitochondria that is associated with α1-AT deficiency-related chronic liver diseases	[444,477,478]
Rat liver tissues liver (I/R and A/R treatments)	Sequestration of GFP-LC3-labeled mitochondria in the liver tissues of I/R- and A/R-treated mice	1. Degradation of dysfunctional mitochondria by mitophagy 2. Ageing aggravated the I/R-induced impairment in Parkin-dependent mitophagy 3. Protection against I/R-induced liver injury by Parkin-dependent mitophagy	[482,483]
1. Liver specimens from human patients 2. Primary mouse hepatocytes 3. Liver tissues of I/R-treated wild type and Sirtuin 1 (SIRT1) knockout mice	1. Electron micrographs showed autophagic vacuoles that engulfed mitochondria in the liver tissues of patients 2. Immunofluorescence staining for tetramethylrhodamine methyl ester (TMRM)-labeled mitochondria in the I/R-treated hepatocytes 3. Degradation of mitochondrial proteins in the I/R-treated hepatocytes 4. Engulfment of MitoTracker-labeled mitochondria by GFP-LC3-labeled autophagic vacuoles in the I/R-treated hepatocytes	Protection against I/R-induced hepatic injury by SIRT1- and Parkin-dependent mitophagy	[483,513,514,515]
Liver tissues of I/R-treated mice	1. Electron micrographs showed autophagic vacuoles that engulfed mitochondria in the liver tissues of I/R-treated mice 2. Degradation of mitochondrial proteins in the I/R-treated hepatocytes 3. Quantification of the mtDNA copy number in the I/R-treated hepatocytes	Protection against I/R-induced liver injury by heme oxygenase-1 (HO-1)-induced mitophagy through phosphoglycerate mutase 5 (PGAM5)	[516]
1. Liver tissues of I/R-treated mice 2. The human normal liver cell line, L02	1. Electron micrographs showed autophagic vacuoles that engulfed mitochondria in the liver tissues of I/R-treated mice 2. Immunofluorescence staining of 5,5′,6,6′-tetrachloro-1,1′,3,3′ –tetraethyl-benzimidazolylcarbocyanine iodide (JC1)-labeled mitochondria in the I/R-treated hepatocytes	Mitophagy induced by the downregulation of microRNA330-3p protected against I/R-induced liver injury by increasing PGAM5 expression	[517]
1. Hepatocytes, Hep3B cell line 2. Primary human hepatocytes (treated with the antiretroviral drug efavirenz)	1. Electron micrographs showed autophagic vacuoles that sequestered mitochondria in the efavirenz-treated hepatocytes 2. Immunofluorescence staining of MitoTracker-labeled mitochondria and LysoTracker-labeled lysosomes in the efavirenz-treated hepatocytes	1. Degradation of damaged mitochondria by efavirenz-induced mitophagy 2. Mitophagy protected against efavirenz-induced hepatic injury	[487,488]
The human normal liver cell line, L02 (cadmium treatment)	1. Electron micrographs showed autophagic vacuoles that engulfed mitochondria in the cadmium-treated liver cells 2. Immunofluorescence staining revealed the co-localization of TOM20-labeled mitochondria with GFP-LC3 in the cadmium-treated liver cells 3. Immunofluorescence staining of MitoTracker- and chloromethyl-X-rosamine (CMXRos)-labeled mitochondria in the cadmium-treated liver cells	1. Degradation of damaged mitochondria by cadmium-induced mitophagy 2. Mitophagy protected against cadmium-induced hepatotoxicity	[489]
Liver tissues from wild type and GFP-LC3 transgenic mice (acetaminophen treatment)	1. Electron micrographs showed autophagic vacuoles that engulfed mitochondria in the acetaminophen-treated mouse hepatocytes 2. Immunofluorescence staining of MitoTracker-labeled mitochondria in the acetaminophen-treated mouse hepatocytes 3. Degradation of mitochondrial proteins in the acetaminophen-treated mouse hepatocytes	1. Degradation of damaged mitochondria by acetaminophen-induced PINK1/Parkin-dependent mitophagy 2. Protection against acetaminophen-induced liver injury by PINK1/Parkin-dependent mitophagy 3. Activation of acetaminophen-induced mitophagy through AMPK activation and the suppression of inflammasome activation	[489,490,491,492,493,494,495]
1. Liver tissues from wild type and GFP-LC3 transgenic mice 2. Isolated primary mouse hepatocytes	1. Electron micrographs showed autophagic vacuoles that engulfed mitochondria in the liver tissues of ethanol-treated mice 2. Engulfment of MitoTracker-labeled mitochondria by GFP-LC3-labeled autophagic vacuoles	1. Degradation of damaged mitochondria by ethanol-induced mitophagy 2. Protection against ethanol-induced liver injury by Parkin-dependent mitophagy	[446,484,485,486]

**Table 3 cells-09-00831-t003:** Summary of the roles of mitophagy in steatosis and fatty liver diseases.

Experimental Model	Characteristics of Mitophagy	Function of Mitophagy	References
Liver tissues and primary hepatocytes from wild type and BNIP3 knockout mice	1. Immunofluorescence staining for Hsp60, a mitochondrial matrix protein, in the primary hepatocytes isolated from wild type and BNIP3-null mice 2. Immunofluorescence staining of MitoTracker-labeled mitochondria in the primary hepatocytes isolated from wild type and BNIP3-null mice	Reduced mitochondrial turnover and increased mitochondrial mass induced by a deficiency in BNIP3-dependent mitophagy	[481]
Liver specimens from patients with alcoholic and nonalcoholic liver diseases (NAFLD)	Electron micrograph of giant mitochondria in the liver specimens of patients with alcoholic and nonalcoholic fatty liver diseases	Mitochondrial dysfunction in the development and pathogenesis of alcoholic and nonalcoholic fatty liver diseases	[526,527,528]
Rat liver tissues (ethanol treatment)	1. Electron micrographs showed autophagic vacuoles that engulfed mitochondria in the ethanol-treated rat liver tissues 2. Immunogold labelling of PINK1 within autophagic vacuoles in which mitochondria were sequestered	1. Degradation of damaged mitochondria by ethanol-induced mitophagy 2. Protection against ethanol-induced fatty liver by PINK1-dependent mitophagy	[529]
Rat liver tissues (ethanol treatment)	1. Electron micrographs showed autophagic vacuoles that engulfed mitochondria in the ethanol-treated rat liver tissues 2. Immunofluorescence staining revealed the co-localization of Parkin and LC3 with mitochondrial and lysosomal markers in the ethanol-treated rat liver tissues 3. Immunogold labelling of Parkin and PINK1 within autophagic vacuoles revealed the sequestration of damaged mitochondria in the ethanol-treated rat liver tissues	1. Degradation of damaged mitochondria by ethanol-induced mitophagy 2. Protection against ethanol-induced fatty liver by PINK1/Parkin-dependent mitophagy	[530,531]
1. Liver tissues from wild type and GFP-LC3 transgenic mice 2. Isolated primary mouse hepatocytes	1. Electron micrographs showed autophagic vacuoles that engulfed mitochondria in the liver tissues of ethanol-treated mice 2. Engulfment of MitoTracker-labeled mitochondria by GFP-LC3-labeled autophagic vacuoles	1. Degradation of damaged mitochondria by ethanol-induced mitophagy 2. Protection against ethanol-induced liver injury by Parkin-dependent mitophagy	[446,484,485,486]
Liver specimens from patients with nonalcoholic liver diseases	Electron micrograph of megamitochondria containing linear crystalline inclusions	Mitochondrial dysfunction in the development and pathogenesis of NAFLD	[533]
Liver tissues from wild type and acyl-CoA:lysocardiolipin acyltransferase-1 (ALCAT1) knockout mice (fed a high-fat diet (HFD))	1. Electron micrographs showed autophagic vacuoles that contained mitochondria in the liver tissues of HFD-fed mice 2. Immunofluorescence staining of MitoTracker-labeled mitochondria in the liver tissues of HFD-fed mice 3. Immunofluorescence staining revealed the co-localization of mitochondria and LC3 with mitochondrial and lysosomal markers in the liver tissues of HFD-fed mice	1. Degradation of damaged mitochondria by HFD-induced mitophagy 2. Protection against HFD-induced NAFLD by Parkin-dependent mitophagy	[534]
1. Human hepatoma cell line, HepG2 cells 2. Mouse liver tissues (fed a methionine and choline-deficient diet (MCD))	1. Electron micrographs showed autophagic vacuoles that contained mitochondria in the thyroid hormone-treated HepG2 cells 2. Immunofluorescence staining for mRFP-EGFP-labeled mitochondria in the thyroid hormone-treated HepG2 cells	1.Induction of mitophagy by thyroid hormone 2. Suppression of NAFLD development by thyroid hormone-induced mitophagy	[535]
Human hepatoma cell line, HepG2 cells (oleic acid (OA) treatment)	1. Immunofluorescence staining revealed the co-localization of MitoTracker-labeled mitochondria and LysoTracker-labeled lysosomes in the OA-treated HepG2 cells 2. Degradation of mitochondrial proteins in the OA-treated HepG2 cells	Activation of DRAM-mediated mitophagy in the progression of NAFLD	[536]
Liver tissues from wild type mice and liver-specific dynamin-related protein 1 (DRP1) knockout mice, optic atrophy protein 1 (OPA1) knockout mice and DRP1/OPA1 double knockout mice (MCD feeding)	1. Immunofluorescence staining revealed anti-pyruvate dehydrogenase E1 (PDH1) antibody-labeled mitochondria in mouse liver tissues 2. Electron micrographs showed depolarized mitochondria in the liver tissues of DRP1, OPA1, and DRP1/OPA1 knockout mice	1. Inhibition of mitophagy during the progression of NAFLD 2. Ubiquitination of mitochondria p62/SQSTM1/Kelch-like ECH-associated protein (KEAP1)/RING-box protein 1 (RBX1) in Parkin-independent mitophagy 3. Induced megamitochondria by impairing Parkin-independent mitophagy in fatty liver	[537]
Liver tissues from wild type and low-density lipoprotein cholesterol receptor (LDLR) knockout mice (Western diet (WD) feeding)	Degradation of mitochondrial proteins in the liver tissues of wild type and LDLR knockout mice	1. Reduced protein stabilities of oxidative phosphorylation subunits in WD-induced NAFLD mice 2. Increased protein degradation of mitochondrial proteins by activated mitophagy	[538]
1. Mouse liver tissues (fed a high-fat/calorie diet (HFCD)) 2. Rat primary hepatocyte (Palmitic acid, PA)	1. Immunofluorescence staining revealed the co-localization of LC3B and cytochrome C oxidase subunit IV in the liver tissues of HFCD-fed mice 2. Induced expression of PINK1 and Parkin in the liver tissues of HFCD-fed mice 3. Immunofluorescence staining revealed the co-localization of LC3 and MitoTracker-labeled mitochondria in PA-treated rat hepatocytes 4. Degradation of TOM20 in PA-treated rat hepatocytes	1. Impaired mitophagy in mice with HFCD-induced NAFLD 2. Induction of inflammasome activation by the suppression of mitophagy in NAFLD mice and primary hepatocytes	[539]

**Table 4 cells-09-00831-t004:** Summary of the roles of mitophagy in liver cancer.

Experimental Model	Characteristics of Mitophagy	Function of Mitophagy	References
1. The BALB/c hepatoma cell line ML-1 2. Liver tissues from NOD/LtSz-PrkdcJ (SCID) mice (treated with concanavalin A (ConA))	1. Electron micrographs showed autophagic vacuoles that engulfed mitochondria in the ConA-treated ML-1 cells 2. Induction of BNIP3 expression in the ConA-treated ML-1 cells 3. Immunofluorescence staining revealed the co-localization of MitoTracker-labeled mitochondria and LysoTracker-labeled lysosomes	1. Activation of BNIP3-dependent mitophagy by ConA treatment 2. Suppression of hepatoma cell growth by ConA-induced mitophagy 3. Inhibition of liver tumor nodule formation by ConA-induced mitophagy	[549,550]
Human hepatoma cell line, HepG2 cells (adriamycin treatment)	1. Electron micrographs showed autophagic vacuoles that engulfed mitochondria in the adriamycin-treated HepG2 cells 2. Immunofluorescence staining of dsRed2-labeled mitochondria and JC1-labeled mitochondria in the adriamycin-treated HepG2 cells	Induction of adriamycin-induced cell apoptosis of human hepatoma cells	[551]
Human hepatoma cell line, HepG2 cells (adriamycin and curcumin treatment)	Immunofluorescence staining of dsRed2-labeled mitochondria and JC1-labeled mitochondria in the adriamycin- and curcumin-treated HepG2 cells	Enhancement of adriamycin-induced cell apoptosis of human hepatoma cells by curcumin	[552]
1. Human hepatoma cell line, HepG2 cells 2. Human hepatoma cell line, Hep3B cells 3. Human hepatoma cell line, Huh7 cells (melatonin and sorafenib treatment)	1. Immunofluorescence staining for anti-TOM20 antibody-labeled mitochondria and anti-LAMP2 antibody-labeled lysosomes in the melatonin- and sorafenib-treated hepatoma cells 2. Degradation of HSP60 in the melatonin- and sorafenib-treated hepatoma cells 3. Decreased mtDNA copy number in the melatonin- and sorafenib-treated hepatoma cells	The cytotoxicity of sorafenib was increased in human hepatoma cells by melatonin-induced mitophagy	[553]
1. Human normal liver cell line, 7702 2. Human hepatoma cell line, HepG2 cells 3. Human hepatoma cell line, Hep3B cells 4. Human hepatoma cell line, Huh7 cells	1. Immunofluorescence staining revealed the co-localization of anti-HSP60 antibody-labeled mitochondria and GFP-LC3-labeled autophagic vacuoles in the nutrient-starved hepatoma cells 2. Mitochondrial translocation of DRAM in the nutrient-starved hepatoma cells	1. The apoptosis of human hepatoma cells was induced by activating DRAM-mediated mitophagy 2. Inhibition of mitophagy-triggered cell apoptosis through the inhibition of the mitochondrial translocation of DRAM	[554]
1. Liver tissues from patients with HCC 2. Liver tissues from wild type and FUNDC1 knockout mice, and liver-specific FUNDC1 knockout mice	1. Degradation of mitochondrial proteins in the liver tumor tissues of patients with HCC 2. Decreased mtDNA copy number in the liver tumor tissues from patients with HCC 3. Induction of FUNDC1 expression in the liver tumor tissues from patients with HCC 4. Electron micrographs showed autophagic vacuoles that engulfed mitochondria in the liver tissues of diethylnitrosamine (DEN)-treated mice 5. Increased the formation of mito-Keima-labeled dot (Red^+^/Green^-^) in the isolated hepatocytes treated by DEN	1. Suppression of the initial development of HCC by FUNDC1-mediated mitophagy through a reduction in inflammasome activation 2. Benefit to tumor growth at the late stage of HCC development by FUNDC1-mediated mitophagy	[555]
1. Liver tissues of HCC patients 2. Human HCC cell lines, Bel7402 and SMMC7721 3. The liver tissues of mouse xenograft models	1. Electron micrograph of deformed mitochondria in the tumorous liver tissues of HCC patients 2. Downregulation of dynamin-1-like protein (DNM1L) expression and upregulation of mitofusin 1 (MFN1) expression in the tumorous liver tissues of HCC patients 3. Co-localization of GFP-LC3B-labeled autophagic vacuoles with MitoTracker red-labeled mitochondria in DNM1L-knockdown Bel7402 and SMMC7721 cells	1. Promotion of HCC cell survival by elevated DNM1L and downregulated MFN1 2. Contribution to poor prognosis of HCC patients by the elevated expressional ratio of DNM1L to MFN1 3. Suppression of tumor growth by inhibition of mitochondrial fission	[556]
1. Human hepatoma cell line, HepG2 cells 2. Human hepatoma cell line, Hep3B cells 3. Human hepatoma cell line, Huh7 cells	1. Immunofluorescence staining revealed the co-localization of nti-TOM20 antibody-labeled mitochondria and phospho-p53 (at serine 392) 2. Phosphorylation of p53 at serine 392 by PINK1 3. Mitochondrial translocation of p53 and p53 phosphorylation in the mitochondria	1. Maintenance of the stemness of cancer stem cells (CSCs) by NANOG gene expression induced by the activation of mitophagy 2. Reduced NANOG gene expression by PINK1-dependent phosphorylation of p53 at serine 392	[557]

**Table 5 cells-09-00831-t005:** Summary of the roles of mitophagy in viral hepatitis.

Experimental Model	Characteristics of Mitophagy	Function of Mitophagy	References
Human hepatoma, Huh7.5.1 cells (hepatitis C virus (HCV) infection)	1. Translocation of Parkin to the mitochondria in HCV-infected cells 2. Mitochondrial ubiquitination was induced by HCV infection 3. Immunofluorescence staining revealed the co-localization of anti-TOM20 antibody-labeled mitochondria, GFP-LC3-labeled autophagic vacuoles, and Parkin in HCV-infected cells 4. Electron micrographs showed autophagic vacuoles that engulfed mitochondria in HCV-infected cells	1. Promotion of virus replication by PINK1/Parkin-dependent mitophagy 2. HCV-induced mitophagy protected infected cells from apoptosis 3. Establishment of viral persistence by HCV-activated mitophagy	[572,574]
Human hepatoma, Huh7.5.1 cells (transfected with HCV NS5A)	1. Immunofluorescence staining for MitoTracker deep red-labeled depolarized mitochondria in HCV NS5A-transfected cells 2. Translocation of Parkin to the mitochondria in HCV NS5A-transfected cells	1. Activation of mitophagy by HCV NS5A through increased ROS production 2. Degradation of depolarized mitochondria by HCV NS5A	[575]
1. Human hepatoma, Huh7 cells (HCV infection) 2. Liver tissues of HCV transgenic mice	1. Decrease in the carbonyl cyanide m-chlorophenyl hydrazone (CCCP)-induced co-localization of anti-TOM20 antibody-labeled mitochondria with Parkin in HCV-infected cells 2. Inhibition of the CCCP-induced mitochondrial translocation of Parkin in HCV-infected cells 3. Interaction of the HCV core protein with Parkin 4. Electron micrographs showed autophagic vacuoles that engulfed mitochondria in CCCP-treated cells, and the change was decreased by HCV infection	A sustained HCV infection induced mitochondrial injury by suppressing mitophagy	[576]
Human hepatoma Huh7 cells (transfection of the HBV viral genome)	1. Translocation of Parkin to the mitochondria in HBV-transfected cells 2. Mitochondrial ubiquitination was induced by HBV transfection 3. Immunofluorescence staining revealed the co-localization of anti-TOM20 antibody-labeled mitochondria, GFP-LC3-labeled autophagic vacuoles, and Parkin in HBV-transfected cells	1. Protection of infected cells from apoptosis by HBV-induced PINK1/Parkin-mediated mitophagy 2. Establishment of viral persistence by HBV-activated PINK1/Parkin-mediated mitophagy	[577]
Human hepatoma cell lines, HepG2 cells, HepG2.2.15 cells, and SMMC-7721 cells (transfection of HBx)	1. Immunofluorescence staining revealed the co-localization of mitochondria, LC3-labeled autophagic vacuoles, and HBx in HBx-transfected cells 2. Induced PINK1 and Parkin expression in HBx-transfected cells 3. Translocation of Parkin to the mitochondria in HBx-transfected cells	Increase in nutrient starvation-induced PINK1/Parkin-dependent mitophagy by HBx	[577]

**Table 6 cells-09-00831-t006:** Summary of the roles of mitophagy in other liver diseases.

Experimental Model	Characteristics of Mitophagy	Function of Mitophagy	References
Rat liver tissues (CCl_4_ treatment)	1. Electron micrographs showed dysfunctional mitochondria in the liver tissues from CCl_4_-treated mice 2. Decreased PINK1 and Parkin expression in the liver tissues from CCl_4_-treated mice	1. Impairment in mitophagy in mice with CCl_4_-induced liver fibrosis 2. Protection against liver fibrosis by melatonin-induced activation of mitophagy	[593]
Mouse liver tissues (CCl_4_ treatment)	1. Electron micrographs showed dysfunctional mitochondria in the Kupffer cells from CCl_4_-treated mice 2. Induced expression of PINK1 and Parkin in the Kupffer cells from CCl_4_-treated mice 3. Increased mitoSOX-labeled mitochondrial ROS levels	1. Activation of PINK1/Parkin-dependent mitophagy in Kupffer cells by CCl_4_-induced liver fibrosis 2. Suppression of PINK1/Parkin-dependent mitophagy in Kupffer cells by T-cell immunoglobulin domain and mucin domain-4 (TIM-4) 3. Mitigation of liver fibrosis by TIM4 interference in Kupffer cells	[594]
1. Human hepatic stellate cell line, LX-2 cells 2. Primary HSCs	1. Degradation of mitochondrial proteins in the PM2.5-treated HSCs 2. Induced expression of PINK1 and Parkin the PM2.5-treated HSCs 3, Mitochondrial translocation of Parkin in the PM2.5-treated HSCs	1. Activation of HSCs and induction of liver fibrosis by PM2.5 2. Induction of mitochondrial damage by PM2.5 through ROS production 3. Induction of PINK1/Parkin-dependent mitophagy 4. Alleviation of PM2.5-induced liver fibrosis through the inhibition of mitophagy	[595]
1. Liver specimens from patients with acute liver failure 2. Mouse liver tissues (lipopolysaccharide (LPS) treatment) 3. Human hepatic stellate cell line, LX-2 cells (H_2_O_2_, LPS, N-acetyl-L-cysteine [NAC], carbonyl cyanide-p-trifluoromethoxyphenylhydrazone [FCCP], or oligomycin treatment)	1. Inhibition of PINK1 expression and upregulation of TOM40 in the HSC model of H_2_O_2_-induced acute liver failure 2. Inhibition of PINK1 expression and upregulation of TOM40 in HSCs from mice with LPS-induced acute liver failure	1. Inhibition of mitophagy by ROS in the HSC model of acute liver failure 2. Promotion of inflammasome activation by impairing mitophagy in the HSC model of acute liver failure	[596]
Hepatocytes isolated from wild type mice, orphan nuclear receptor subfamily 4 group A member 1 (NR4A1) knockout mice, and liver-specific DNA-dependent protein kinase catalytic subunit (DNA-PKcs) knockout mice (ethanol treatment)	1. Electron micrographs showed damaged mitochondria in the ethanol-treated hepatocytes 2. Immunofluorescence staining for MitoTracker-labeled mitochondria in the ethanol-treated hepatocytes 3. Degradation of mitochondrial proteins in the ethanol-treated hepatocytes	1. The NR4A1/DNA-PKcs/p53 axis enhanced the pathogenesis of alcohol-related liver disease (ARLD) 2. ARLD pathogenesis was induced by activating DRP1-related mitochondrial fission and restricting FUNDC1-required mitophagy.	[597]
1. Liver tissues from wild type and BNIP3 knockout mice 2. Primary hepatocytes isolated from wild type and BNIP3-null mice	1. Immunofluorescence staining for HSP60 2. Degradation of mitochondrial proteins	1. Increase in lipogenesis via the upregulation of lipogenic enzymes through defective mitophagy 2. Suppression of the β-oxidation of fatty acids by interference with mitophagy	[481]
1. Liver tissues from wild type and thyroid hormone receptor knockout mice 2. HepG2 cells	1. Electron micrographs showed damaged mitochondria 2. Degradation of mitochondrial proteins	Increased β-oxidation of fatty acids through inducing gene expression CPT1α by thyroid hormone-activated mitophagy	[598]
Liver tissues from wild type and REDD1 knockout mice (HFD treatment)	1. Electron micrographs showed damaged mitochondria 2. Degradation of mitochondrial proteins	Increased CPT1α, BNIIP3 and Parkin expression in the livers of HFD-fed REDD1 KO mice	[599]
1. Liver tissues from wild type mice 2. Primary mouse hepatocytes	Co-localization of GFP-LC3 and mitochondria	Suppression of mitophagy by insulin resistance	[600]
1. Liver tissues from wild type and Parkin knockout mice 2. Primary mouse hepatocytes (HFD treatment)	Degradation of mitochondrial proteins	No significant changes in obesity and insulin resistance were observed in response to an impairment in Parkin-dependent mitophagy	[601,602]
1. Liver tissues from wild type and FUNDC1 knockout mice 2. Primary mouse hepatocytes (HFD treatment)	1. Electron micrographs showed damaged mitochondria 2. Degradation of mitochondrial proteins 3. Change in the fluorescence intensity of the mt-Keima reporter	Induction of adipose tissue-associated macrophage infiltration and hyperactivation of MAPK8 (also named JNK1) by the loss of FUNDC1-mediated mitochondrial turnover	[603]

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
