# Peer review of "Mitophagy in the Pathogenesis of Liver Diseases"

_cells, 2020, doi:10.3390/cells9040831_

Round 1

Reviewer 1 Report

Dear Author,

your manuscript "Mitophagy in the pathogenesis of liver diseases" describes the contrasting roles of mitochondria autophagy (mitophagy) in the onset of several liver diseases. 

The manuscript is very well written and structured in all the parts. The topic is well explained and many details on molecular mechanisms are presented. Also tables are very detailed and figures make clearer the autophagy molecular mechanisms. 

I suggest only two corrections about typing errors:

  • page 5, paragraph 3.2: eliminate the underline for "successively";
  • page 7, paragraph 4.2: eliminate the underline for "PINK304-305. Stabilization of PINK1 leads to its accumulation on the".

Author Response

Dear reviewer:

Thank you for giving me the opportunity to resubmit my manuscript entitled “Mitophagy in the pathogenesis of liver diseases” to Cells (Manuscript ID: cells-749271). I appreciate for the thoughtful and constructive suggestions provided by the reviewers. The content of this manuscript has been improved based on the reviewers’ comments, and I have incorporated sections introducing the role of autophagy in liver physiology and liver diseases, the role of mitophagy in regulating hepatic insulin resistance and fatty acid oxidation and a discussion of the potential use of mitophagy modulators as a curative therapy for liver diseases. The changes are shown in the revised manuscript, and point-by-point responses to each comment are listed below.

Point 1: Dear Author, your manuscript "Mitophagy in the pathogenesis of liver diseases" describes the contrasting roles of mitochondria autophagy (mitophagy) in the onset of several liver diseases. The manuscript is very well written and structured in all the parts. The topic is well explained and many details on molecular mechanisms are presented. Also tables are very detailed and figures make clearer the autophagy molecular mechanisms.

Response 1: I appreciate the reviewer’s recognition of my efforts in writing this review manuscript and recommendation for the publication of this review article in Cells.

Point 2: I suggest only two corrections about typing errors: page 5, paragraph 3.2: eliminate the underline for "successively"; page 7, paragraph 4.2: eliminate the underline for "PINK304-305. Stabilization of PINK1 leads to its accumulation on the".

Response 2: Thank you for these comments. I apologize for these mistakes in formatting and these mistakes have been corrected. The unusual underlined characters in the text have been removed. Please see line 4 of paragraph 2 on page 5 (successfully), line 9 of paragraph 3 on page 5 (Subsequently), and lines 9~10 of paragraph 2 on page 7 (Mitochondrial import interferes….; Stabilization of PINK1 leads to its accumulation on the…) in the revised manuscript.

We hope that this version of our manuscript and our responses address all your concerns and that this revised manuscript meets the criteria for publication in Cells. Thank you for your kind consideration.

Reviewer 2 Report

This is an extensive and dense review with over 600 references .Overall it is very comprehensive and thorough.

I have the following comments

1.I think there should be some reference to autophagy and the liver given parts of the review give a general review on autophagy

2.Perhaps the general over view of autophagy and mitophagy can be shortened ( hundreds of references here)

2.I think the written word is too dense and there are very long paragraphs that need breaking up

3.The tables contain much of the information that is in the text .The tables should be shortened to just a list of studies and outcomes

Author Response

Dear reviewer:

Thank you for giving me the opportunity to resubmit my manuscript entitled “Mitophagy in the pathogenesis of liver diseases” to Cells (Manuscript ID: cells-749271). I appreciate for the thoughtful and constructive suggestions provided by the reviewers. The content of this manuscript has been improved based on the reviewers’ comments, and I have incorporated sections introducing the role of autophagy in liver physiology and liver diseases, the role of mitophagy in regulating hepatic insulin resistance and fatty acid oxidation and a discussion of the potential use of mitophagy modulators as a curative therapy for liver diseases. The changes are shown in the revised manuscript, and point-by-point responses to each comment are listed below.

Point 1: I think there should be some reference to autophagy and the liver given parts of the review give a general review on autophagy.

Response 1: I am very grateful for the reviewer’s thoughtful comment on the discussion and references related to autophagy in liver function and disease progression. We have extended our manuscript to include a section (section 5) describing the roles of autophagy in regulating liver function and liver diseases in the revised manuscript. Please see lines 1-26 of paragraph 2, section 5 on page 11 in the revised manuscript.

Point 2: Perhaps the general over view of autophagy and mitophagy can be shortened (hundreds of references here).

Response 2: Thank you for these comments. I am very grateful to reviewer’s thoughtful suggestions on the section providing an overview of autophagy and mitophagy. In the past few decades, numerous studies proposed new hypotheses, concepts, and conclusions about the regulation of autophagy and mitophagy and the roles of autophagy and mitophagy in liver diseases. Previous review articles often simply focused on the introduction of the process of autophagy and the function of autophagy in liver diseases. The detailed molecular mechanism underlying autophagy was usually omitted in these papers. To date, a comprehensive review of the functional role(s) of mitophagy in liver diseases has not published in the literature. In addition, numerous new mechanisms for the regulation of autophagy and mitophagy have been identified in recent years. Most importantly, the discovery of autophagy and mitophagy was mostly related to liver physiology and pathology. Therefore, I intend to provide compelling information about the history of the discovery of autophagy and mitophagy, the molecular mechanisms of autophagy and mitophagy, the molecular mechanisms regulating autophagy and mitophagy pathways, and new findings from autophagy and mitophagy research in this review article. This comprehensive knowledge will help readers to understand the regulation of autophagy and mitophagy (including the compelling regulators of autophagy and mitophagy), thus allowing readers to understand how autophagy and mitophagy are involved in the regulation of liver physiology and the development of liver diseases. The detailed information provided in this review article will likely inspire the research community to develop new ideas for the study of mitophagy in liver disease, and prompt us to search and/or identify new strategies to target mitophagy for curing liver diseases. I hope that the reviewer kindly agrees with me to keep the completeness of the autophagy and mitophagy overview in the revised manuscript. Thank you again for the constructive comments.

Point 3: I think the written word is too dense and there are very long paragraphs that need breaking up.

Response 3: We appreciate the reviewer’s suggestions and have revised manuscript. The long paragraphs have been revised and separated into two paragraphs in the revised manuscript. Additionally, new titles for separate sections have been added to the revised manuscript. Please see lines 1-2, paragraph 2, sections 2.3 and 2.3.1 on page 3; line 1, paragraph 3, section 2.3.2 on page 3; line 1, paragraph 2, section 2.3.3 in page 4; line 1, paragraph 3, section 2.3.4 on page 4; the separation of two paragraphs in Section 4.1 on page 6; the separation of two paragraphs in section 4.2 on page 7; the separation of five paragraphs in section 4.3 on pages 7-9; the separation of two paragraphs in section 6.2 on page 12; the separation of two paragraphs in section 7.2 on pages 13-14; the separation of three paragraphs in section 7.5 on page 16 in the revised manuscript. Thank you again for your constructive comments. 

Point 4: The tables contain much of the information that is in the text. The tables should be shortened to just a list of studies and outcomes.

Response 4: Thank you for these comments. I have revised the content of each tables to concisely illustrate the studies and outcomes of different papers. Please see tables 1~6 on pages 55~64 in the revised manuscript. Numerous studies have shown the modulation of mitophagy and the potential role(s) of mitophagy in the development of liver diseases. However, large discrepancies and controversial conclusions were obtained by different studies, which may have been related to the use of different types of approaches, cell contexts, and experimental settings. Some studies analyzed the regulation and function of mitophagy in the context of cell culture models in vitro, whereas other groups utilized small animal models, such as mouse rats, and/or liver specimens from human patients. Additionally, different models of liver diseases, including different stimuli-induced liver diseases used in different studies, also led to divergent and complicated conclusions on the regulation of mitophagy in liver diseases. In addition, mitophagy was shown to affect different stages of liver diseases and lead to different outcomes during disease progression. Therefore, I intend to provide compelling information on the experimental model, the setting of disease modeling, the detection of mitophagy, the major conclusion about the role of mitophagy in liver diseases, and the related references. I believe that the summarization of this information in tables may allow readers to clearly view how and/or why different conclusions were obtained from different studies. I hope that the reviewer agrees with the preservation of the comprehensive information in each tables. Thank you again for your constructive comments.    

We hope that this version of our manuscript and our responses address all your concerns and that this revised manuscript meets the criteria for publication in Cells. Thank you for your kind consideration.

Reviewer 3 Report

Review for cells-749271

Several other mechanisms have been reported earlier. In this review, does the author wish to summarize in this review and attract great interest from new readers?

  • Most of the author's descriptions are lengthy, and topics without major breakthroughs should be more concise. I suggest that the authors reorganize and review on new research on mitophagy in liver studies in recent years.

  • The authors describe the involvement of mitophagy in liver disease, but have not discussed any treatments and molecular regulation of hepatic insulin resistance, fatty acid oxidation. The authors should improve the comments in this section.

  • In addition, Figure 1 is very similar to the image published by the author on Int J Mol Sci. (2019 Jan 13; 20 (2). Pii: E300.). Are there any doubts about self-plagiarism?

Author Response

Dear reviewer:

Thank you for giving me the opportunity to resubmit my manuscript entitled “Mitophagy in the pathogenesis of liver diseases” to Cells (Manuscript ID: cells-749271). I appreciate for the thoughtful and constructive suggestions provided by the reviewers. The content of this manuscript has been improved based on the reviewers’ comments, and I have incorporated sections introducing the role of autophagy in liver physiology and liver diseases, the role of mitophagy in regulating hepatic insulin resistance and fatty acid oxidation and a discussion of the potential use of mitophagy modulators as a curative therapy for liver diseases. The changes are shown in the revised manuscript, and point-by-point responses to each comment are listed below.

Point 1: Several other mechanisms have been reported earlier. In this review, does the author wish to summarize in this review and attract great interest from new readers?

Response 1: I am very grateful for the reviewer’s thoughtful comment on this manuscript. I completely agree with the reviewer’s concern regarding the other mechanism reported in previous studies, such as DNA damage-induced mutations in oncogenes and/or tumor suppressor genes and alterations in cell growth signaling that may potentially regulate the development of liver diseases. The related knowledge of these factors involved in liver disease progression has been extensively discussed in several critical review articles. Although the role and regulation of autophagy may be summarized in other review papers, a comprehensive overview of the regulation of selective autophagy, particularly mitophagy, has not been reviewed in academic journals. Based on accumulating evidence, mitophagy promotes the turnover of mitochondria in live cells to support the regulation of metabolism for liver physiology, and a protective role for mitophagy has emerged in the prevention of liver disease development. In contrast, mitophagy was also shown to be involved in the development and progression of liver-related diseases. Collectively, mitophagy is likely differentially regulated in different types and at different stages of liver-related diseases. The functional role(s) of mitophagy in liver diseases could depend on the context and disease stage. Nevertheless, approaches targeting mitophagy have become a feasible strategy to design new therapies as an intervention for liver diseases. However, whether and how mitophagy functions in the prevention and/or promotion of liver-associated diseases are still topics of debate due to the large discrepancies among different studies. In addition, new findings and new concepts regarding the interaction between mitophagy and the liver are still being reported. Therefore, an updated review article describing the detailed molecular mechanism of mitophagy, the functions of mitophagy in liver physiology, the modulation of mitophagy during the development of liver diseases, the involvement of mitophagy in the progression of liver diseases, and the therapeutic potential of mitophagy manipulations for curing liver diseases in the clinic is urgently needed for research community. I also believe that this review article meets this requirement and hope that the reviewer agrees with the publication of my revised manuscript. Thank you again for the thoughtful suggestions.

Point 2: Most of the author's descriptions are lengthy, and topics without major breakthroughs should be more concise. I suggest that the authors reorganize and review on new research on mitophagy in liver studies in recent years.

Response 2: Thank you for these comments. I am very grateful to the reviewer for the thoughtful suggestions on this manuscript. We have revised and reformatted this manuscript. Several long paragraphs have been separated and many sub-sections have been added to the revised manuscript. Please see lines 1-2, paragraph 2, sections 2.3 and 2.3.1 on page 3; line 1, paragraph 3, section 2.3.2 on page 3; line 1, paragraph 2, section 2.3.3 in page 4; line 1, paragraph 3, section 2.3.4 on page 4; the separation of two paragraphs in Section 4.1 on page 6; the separation of two paragraphs in section 4.2 on page 7; the separation of five paragraphs in section 4.3 on pages 7-9; the separation of two paragraphs in section 6.2 on page 12; the separation of two paragraphs in section 7.2 on pages 13-14; the separation of three paragraphs in section 7.5 on page 16 in the revised manuscript. I appreciate the reviewer’s suggestions on the precise and simply constructed overview of mitophagy in liver diseases from recent studies. Because the interaction between mitophagy and liver diseases has not reviewed before, I intend to provide a comprehensive review of the discovery of mitophagy in the liver and the regulation of mitophagy in the liver, and historically summarize the functional role(s) of mitophagy in liver function and the contribution of altered mitophagy to the pathogenesis of liver diseases. The compelling information not only provides readers the opportunity to understand all the possible regulatory mechanisms and role(s) of mitophagy in liver diseases but also may inspire the readers to compare the discrepancies between different studies using different experimental settings and eventually identify a new and conceivable model for further investigations on mitophagy and liver diseases. I hope that the reviewer will agree with the preservation of original title and content I intended to present in this manuscript. Thank you again for the constructive comments.

Point 3: The authors describe the involvement of mitophagy in liver disease, but have not discussed any treatments and molecular regulation of hepatic insulin resistance, fatty acid oxidation. The authors should improve the comments in this section.

Response 3: I appreciate the reviewer’s suggestions and have revised manuscript. The content regarding the regulation of mitophagy in hepatic insulin resistance and fatty acid oxidation has been incorporated into the revised manuscript. Please see section 7.5, lines 1-26, paragraphs 2 and 3 on page 16 in the revised manuscript. The discussion on the therapeutic potential of mitophagy modulators in treating liver diseases have been improved and strengthened in the revised manuscript. Please see section 8, page 16, and paragraphs 1 and 2 on page 17 of the revised manuscript. Thank you again for the constructive suggestions.

Point 4: In addition, Figure 1 is very similar to the image published by the author on Int J Mol Sci. (2019 Jan 13; 20 (2). Pii: E300.). Are there any doubts about self-plagiarism?

Response 4: Thank you for these comments. I apologize for this mistake and the potential concern regarding the similar presentation in two articles related to the negligence in the preparation of submission. I have completely prepared new Figures 1 and 2 (pages 67~69 in the revised manuscript) and their legends (pages 65~66 in the revised manuscript) for publication in the revised manuscript. The contents of Figures 1 and 2 have been described more detail to fit the information presented in this revised paper. In Figure 1 of revised manuscript, the molecular processes of microautophagy (A), chaperone-mediated autophagy (B), and macroautophagy (C) are separately presented. Detailed descriptions of the membrane resource of phagophore formation, ATG9-, VMP1, and COPII-containing vesicles, the involvement of ESCRT in microautophagy, and the translocation of the LAMP2 complex for chaperone-mediated autophagy have been incorporated. In Figure 2 of revised manuscript, two distinct processes for cargo receptor- and additional adaptor-mediated selective autophagy are presented in (A). In Figure 2B of the revised manuscript, the schematic shows different types of selective autophagy and the information about the cargo receptors of different kinds of selective autophagy has been summarized in a table. Please see Figures 1 and 2 of the revised manuscript. Thank you again for your comments.

We hope that this version of our manuscript and our responses address all your concerns and that this revised manuscript meets the criteria for publication in Cells. Thank you for your kind consideration.

Round 2

Reviewer 2 Report

none 

Reviewer 3 Report

The author has address all my concerns.
No further comments.